# Artificial Neural Network Blockchain Techniques for Healthcare System: Focusing on the Personal Health Records

**Seong-Kyu Kim [1] and Jun-Ho Huh [2,\*]**

1   Department of Information Technology, Sungkyunkwan University, Seoul 03063, Korea; guitara7@skku.edu
2   Department of Data Informatics, Korea Maritime and Ocean University, Busan 49112, Korea
\*   Correspondence: 72networks@kmou.ac.kr; Tel.: +82-51-410-4371

**Abstract:** This paper seeks to use artificial intelligence blockchain algorithms to ensure safe verification of medical institution PHR data and accurate verification of medical data as existing vulnerabilities. Artificial intelligence has recently spread and has led to research on many technologies thanks to the Fourth Industrial Revolution. This is a very important factor in healthcare as well as the healthcare industry's position. Likewise, blockchain is very safe to apply because it encrypts and verifies these medical data in case they are hacked or leaked. These technologies are considered very important. This study raises the problems of these artificial intelligence blockchains and recognizes blockchain, artificial intelligence, neural networks, healthcare, etc.; these problems clearly exist, so systems like EHR are not being used. In the future, ensuring privacy will be made easier when these EHRs are activated and data transmission and data verification between hospitals are completed. To overcome these shortcomings, we define an information security blockchain artificial intelligence framework and verify blockchain systems for accurate extraction, storage, and verification of data. In addition, various verification and performance evaluation indicators are set to obtain the TPS of medical data and for the implementation of standardization work in the future. This paper seeks to maximize the confidentiality of blockchain and the sensitivity and availability of artificial intelligence.

**Keywords:** blockchain; artificial neural network blockchain; artificial intelligence; intelligent agent healthcare; personal health record; PHR

## 1. Introduction

This study aimed to produce many synergistic effects via the healthcare system using a neural network and blockchain, the latest Fourth Industrial Revolution technology that has been widely applied recently. There is no technology that has received as much attention as artificial intelligence and blockchain. There are many pros and cons, not to mention the many technical issues to be solved, but artificial intelligence and blockchain are expected to revolutionize many areas after institutional overhaul and trial and error. Artificial intelligence and blockchain have great connections with healthcare as well as finance and logistics. Data is the basis of recent healthcare innovations. Advances in digital technology have changed medical data quantitatively and qualitatively, from genetic information to medical records. These changes require enhanced ownership and security, privacy, integrity, and traceability in the production, storage, transmission, sharing, and utilization of medical data. Artificial intelligence blockchain is a technology that can store data and prevent forgery, so it is likely to meet the new needs of these medical data. While artificial intelligence and blockchain are available in the medical sector, ranging from clinical trial data management of new drug development to medical insurance screening, the first aspect currently being investigated is that medical systems

to overcome traditional medical markets have long been established, using artificial intelligence and blockchain in medical data such as medical records. Currently, medical, examination, prescription, and diagnostic data for patients are logged in the records of each hospital's electronic server or in a paper chart. Thanks to blockchain, data can be more reliable, secure, interoperable, and accessible. Medical blockchains in the US and MediBlock in Korea are pursuing this purpose. The second aspect is to use artificial intelligence and blockchain to utilize or trade medical data in a decentralized manner without the patient himself being an intermediary vendor. Patients who have already sold subscriber data and provided actual data, even if middle-class vendors made profits, were not given the right to decide or the benefits of trading.

For example, the hospital information service provider's subscriber genetic information is sold to pharmaceutical companies to generate revenue, while the patient community generates revenue by selling data on the side effects of anonymous patients. On the other hand, patients who provided the information had less say in decision making and received no profit in sales. In addition, using artificial intelligence and blockchain can reduce these middle-market roles and allow direct decisions to enable patients to utilize their data and increase profits. In this case, you decide how much your data will be sold to which company's projects, and you can track the post-sale usage process. Incentives will make patients participate more actively, which also benefits the data buyer. The purpose is to encourage people to take good care of their health and to utilize it. Encouraging people to exercise, quit smoking, quit drinking, and control drug intake is a big challenge for the medical community. Medical information systems and healthcare systems that are inextricably linked to artificial intelligence and blockchain can be applied to and may be covered by insurance.

It also requires multidisciplinary collaboration among computer scientists, engineers and data scientists as well as statisticians and other stakeholders to make the most of the potential of big data. There is also a need to enhance decision-making and service delivery by improving big data processing, management and analysis infrastructure through large-scale investment and development by businesses and other organizations. Moreover, there is an urgent need to integrate new big data applications with existing APIs (Application Programming Interfaces) such as SQL (Structured Query Language) and R language for statistical computing. Several leading IT companies such as IBM, Oracle, Microsoft, SAP and HP have already invested more than $15 billion in big data systems. One of the challenges big data faces in enterprise and cloud infrastructure is the presence of tenants with different SLA (Service-Level Agreement) requirements and various workloads that must be hosted on the same set of clusters. The initial solution to these challenges at the application level is to leverage distributed file systems to control data access and sharing within the cluster. At the infrastructure level, solutions such as VM (Virtual Machines) or Linux containers dedicated to each application or tenant enable the separation of allocated resources. Big data systems are also plagued by security, privacy and governance concerns. It is also necessary to investigate and optimize energy-efficient processing and networking infrastructure for future big data, as the growing computing demand for data volumes exceeds the capabilities of existing commercial infrastructure [1–3].

Now, as we have changed the industrial system based on convergence with the technologies of the Fourth Industrial Revolution, we have created a new way of life and approach in the present era, and healthcare-based industries are also changing. Smart healthcare is changing the way medical data is accessed and secured through convergence with healthcare and patients using converged technology medical device services. Based on the development of various types of services, such as disease management, healthcare, and dietary management, customized treatment services such as precision medicine, telemedicine monitoring services, and smart healthcare services are expected to be expanded to manage the entire life cycle comprehensively. Smart device technologies and other technologies are integrated with healthcare and medical services, consisting of software companies, hardware manufacturers, services, businesses, government departments, and regulations. Software companies analyze the data collected from healthcare-related application platform systems, etc., and include medical and health information solutions and personal health record solution-based analysis tool

platforms. To collect healthcare-related data, the hardware manufacturing industry manufactures and collects data such as hardware wearable smart devices and has personal health devices, wellness devices, and communication devices sensors. Service companies are those that provide personalized healthcare and medical services for each patient user, including health information and analysis services, personalized healthcare services, telemedicine, etc., as models that operate around hospitals and health centers. This empirical study proposes to use a neural network for accurately identifying customer information and providing personalized medical information using blockchain to verify that they are appropriate users. The rest of this paper is organized as follows: Section 1 describes the healthcare services that combine artificial intelligence and blockchain; Section 2 examines related studies: neural computing, blockchain, and deep learning; Section 3 explores the artificial intelligence and blockchain healthcare framework as the actual application methodology; Section 3 also describes the actual implementation model; finally, Section 4 presents the conclusion and future challenges (see Figure 1).

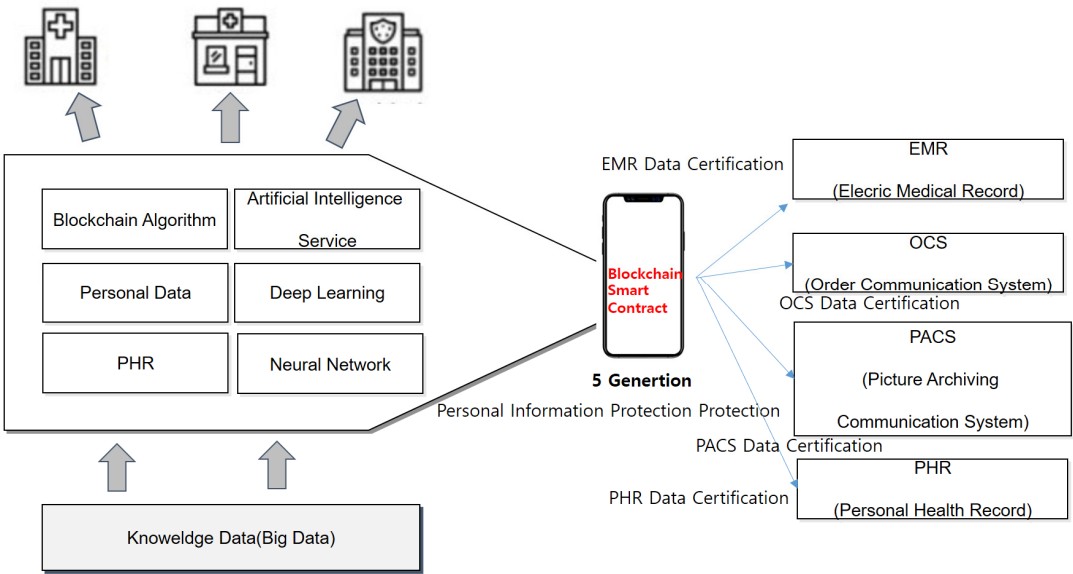

**Figure 1.** Healthcare verification concepts with blockchain and artificial intelligence.

This paper also designs image data for medical data in EMR (Electric Medical Record), PACS (Picture Archiving Communication System), and OCS (Order Communication System) and PHR (Personal Health Record) to match the actual various medical data. The reason is that medical data can currently only be seen by doctors, but they use medical terminology. In this paper, the first issue is to eliminate forgery and falsification of medical data. Secondly, medical data is important personal information. Because it is sensitive information, medical data are regulated very strongly in Europe as well as in Korea, via what is called GDPR (General Data Protection Regulation). The purpose of this important medical data regulation is to prevent forgery and to help personal medical data flow.

## 2. Background Knowledge

### 2.1. Artificial Intelligence Neural Network

An artificial neural network is a model of machine learning modeled after the structure of a human neural network. As the name suggests, it is an algorithm modeled after the neural network of living things, especially human visual/auditory cortex. It is about a million light years away from how the real human brain works. It has not been such a promising field from the start. For a while, it was almost run by SVMs (Support Vector Machines) and NBCs (Neumann Boundary Conditions), an unusual situation that was unexpectedly remedied by some scientists who had been silently studying it for

decades, despite such cynicism. The basic principle is simple. Create several layers and put "cells" in them, and connect them with random connections. The "cells" each multiply the signals that come into them by their weight and add them all to "neural", then deliver the signals to the next neuron (wx + bwx + bwx + b) relative to its threshold. If you do this, the transmitted signal is just the linear sum of the input signal; no matter how complex, the linear sum is repeated, all you get is the linear sum of the input signal. In other words, no matter how many layers you stack, you get the same result as when you multiply the matrix once.

Ethereum is stored as an additional ledger-data-called DLT (Distributed Ledger Technology) system featuring a validating machine, along with an additional ledger-data-called DLT system featuring a validating machine. Like Bitcoin, it offers similar functions. In addition, Ethereum's EVM allows smart contracts to be placed and executed in public books, enabling the creation of immutable computer logic. Smart contract introduction and execution of smart contracts require Etherithostore's accrual of authority by spending a certain amount of money called Ethereum cryptocurrency. Once placed in the ledger, the simple model of smart contract execution is as follows. Smart contracts run with some input data through transactions. An EVM run uses input data to conclude a mart contract and generate output. This action changes the state of the EVM stored in the ledger along with the output data. The PoW consensus algorithm ensures that the updated status is accurately recorded on all nodes in the network. The open ledger ensures that the transfer of currencies through transactions and changes in the status of EVMs are fully transparent and verified by all participants [4,5].

As mentioned earlier, the linear sum of the input signals is put into the enable function, and the strength of the signal is finally calculated using the best-performing function by simply using the rule-of-thumb function. In the beginning, logistic models were used with sigmoid functions that were likely to simulate the actual working rate of cells, and hospital or medical company functions were used. If the function was used as an active function, there was a problem of the value of the active function differential being multiplied by the value of the active function differential when learning of the strength of the connection was increased, resulting in poor learning. As such, nowadays, functions such as the output value increase as the input value increases, such as in RELU (Rectified Linear Unit), rather than use a squashing function such as sigmoid or tanh. The RELU function has the shape of max (0, x); if the input value is less than 0, there is a problem, i.e., the learning of the neuron does not proceed for the negative input value. To address this, leaky RELU functions and parametric RELU functions with non-zero output values have been proposed even at negative input values and are generally known to perform better than RELU.

In addition, an RELU function with continuous differential values in the function is also widely used as an active function. Learning of artificial neural networks will result in updating weights and thresholds in the direction that is expected to minimize errors with learning data. In fact, although described as a threshold, this value can be seen as the strength of the connection with a "bias neuron" with output value of 1, so the threshold can also be virtually seen as a kind of connection strength, or weight. In the course of learning, the weights are updated, not significantly at one time but several times in small increments. This is because weight update directions that are likely to reduce errors are often inefficient. For example, if you walk to a destination with your eyes closed, and you open your eyes once in a while to get a sense of the situation around you, then you are in a situation where you have to close your eyes and move again. Even if you open your eyes once and understand what is going on around you, if you walk too many streets with your eyes closed, you can run into a wall, or you can figure in an accident. On the other hand, if you walk a short distance, open your eyes again to understand the surrounding situation, and walk a short distance again, you may not figure in an accident, but it may take too long to reach your destination. Therefore, walking the right distance is an important issue [6–8].

Similarly, if weights are not updated with appropriate learning rates when learning an artificial neural network, problems arise such as releasing artificial neural networks or taking too long to learn. As the word itself implies, an artificial neural network differs greatly from a biological brain.

The biological brain, especially the human brain, is composed of more than 100 billion cells, and technology so far cannot simulate such a number of neurons. Artificial neural networks are neural networks that humans have created and are used for deep learning. Therefore, we have created a deep learning network like a neuron's neural network. A stronger stimulus does not increase the magnitude of the response, but increases instead the frequency of the response, which in fact can be seen as similar to a step function or a de-lock delta function. For stair functions such as unit stair functions, there has been a very big problem of not being able to differentiate them, because they were discontinuous in places where x = 0x = 0x = 0x = 0x = 0x = 0.

Therefore, scientists and engineers at the time used the sigmoid function and the tahn function, a soft step function that is capable of differentiation. Although the sigmoid function is a function very similar to the step function, there is a big difference from the biological brain, which cannot operate without the passage of time. On the other hand, for artificial neural networks, the input value is given regardless of the time flow, and the output value is set. Moreover, the artificial neural network model itself has evolved, and functions of the RELU family, not the sigmoid function, have become widely used as active functions, which are somewhat different from the step function: hence the considerable difference from the biological brain.

Of course, if you think that the function of the RELU family represents the frequency of the response to the stimulus, then the distance has been narrowed. The problem is that researchers in the past thought that artificial neural networks, which differ from the biological brain, would have difficulty calculating as fast computers because of their quick computing performance, and the artificial neural network model almost faded into oblivion. Of course, such indifference played a crucial role not only in terms of computer performance but also in the fact that, the more hidden layers there are, the more the problem of learning remains unsolved. This artificial neural network is the core technology of artificial intelligence made with a human neural structure (see Figure 2).

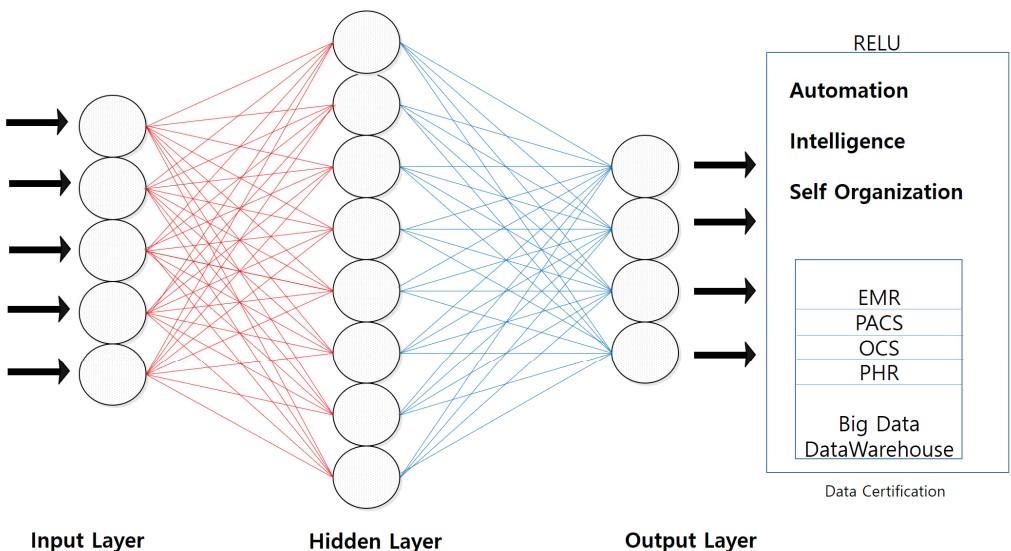

**Figure 2.** Concept of an artificial intelligence neural network for healthcare.

There is a lot of data processed using artistic neural networks. Recently, medical data is also being stored in many data warehouses. Therefore, medical imaging data is being analyzed with blockchain using these data warehouses. This plays a role in verifying using blockchain agreement algorithms in the middle. This paper uses these blockchain verification algorithms and medical data uses artificial intelligence neural networks. Therefore, it is a safer way for patients to determine their personal information's self-controlling rights. These EMR, PACS, OCS, and PHR data are highly sensitive data, so no errors should occur in validating medical data. It is important to study algorithms to overcome these points.

*2.2. Deep Learning*

Deep learning is one of the oldest concepts. In the research phase of the study of artificial intelligence systems, deep learning consists of in-depth practices after machine learning. This is an alternative method when x and y are in a functional relationship, but no model is available to predict y from x. Think of it as a high-compatibility method for regression analysis, an easily understandable concept. This study mainly deals with artificial neural network techniques that stack and connect artificial neurons between input and output [9–12].

The concept is that, if you stack multiple neural networks, deep learning, and RNNs (Recurrent Neural Networks) in different layers, you get a formula-like deep RNN. The artificial neural network itself has been around for quite some time, and neural network models such as CNNs (Convolutional Neural Networks) and RNN have also been actively studied since the 1980s. Moreover, because the environment makes it difficult to use with computers, model deployment was impossible and was considered only theoretical for decades. It was only in the 2000s that those theories became reality [13–15].

Of course, computers were not the only ones that developed for about 20 years. Since the DBN (Deep Belief Network) was announced in 2006, it has been revered as a savior in the artificial intelligence field since it started to receive serious attention in 2009. For example, Facebook's deep-learning-based face recognition model has recorded more than 97 percent, little different from the performance of humans. In object recognition competitions, the NCS (Collaboration Neural Network), a type of deep learning, has outranked all of its long-standing object-recognition algorithms, with no knowledge of any improvement in speech recognition. Google also uses this to create artificial intelligence to learn how to play games on its own or to create a machine that uses search results to learn the concept and appearance of cats. In Korea, Korean companies such as Naver and Kakao are also actively studying deep learning. The biggest feature of deep learning is that it will increase the volume of the model; when data is put into it, it will definitely improve performance. There was a session on deep learning on Naver's Deview 2013, which compared the days before and after deep learning to the Bronze Age and the Iron Age, respectively. Some people tend to mistake deep learning as a magic technique, which is very different from other machine learning; in fact, it is a kind of machine learning algorithm [16]. In other words, the concept used in the same class as deep learning is machine learning. For example, SVMs, which were most popular before deep learning, ended up using the hinge loss function + Frovenius norm regularization as a single layer perceptron without activation, not to mention the logistic registry. In addition, as mentioned before, artificial neural network techniques are statistically a kind of very large and complex hybrid regression model. One of the most problematic issues about deep learning compared to a typical MLP is the use of multiple hidden layers. To explain briefly what this means, there is a form of correction process that lets you get the right answer across every floor. Likewise, because the calibration process continues to multiply and modify numbers from 1 to 0, there is a problem of the slope of the formula gradually approaching zero if it is a deep hidden layer. For this reason, the problem of slope extinction has once been avoided by the artificial neural network academe. Fortunately, several solutions are now largely available, with Bold treatment often applied to the method. The first is batch normalization. To accomplish this, normalize the value from −1 to 1, and then perform ax + b (a and b variables). Originally designed to allow the slope to flow, this quickens learning and also has a generalizing effect. The second is to change the transfer function, which is based on backpropagation that feeds back the error (y − y*) derived by substituting the result (yo*) with yo − y* to the matrix (x1 to xn) [17–20].

In addition, if you add a calculation that skips a layer, the slope may flow along that path. The next step is to add one of the most commonly used connections. Literally, $f'(x) = f(x) + x$. Addition is said to be possible because it lets the slope flow away [21–23]. In addition to the role of letting the slope flow, there is also the added effect of accelerating convergence by allowing the model to learn the error of the value, not the value itself [24]. Using this method, ResNet also piled up 150 layers (see Figure 3).

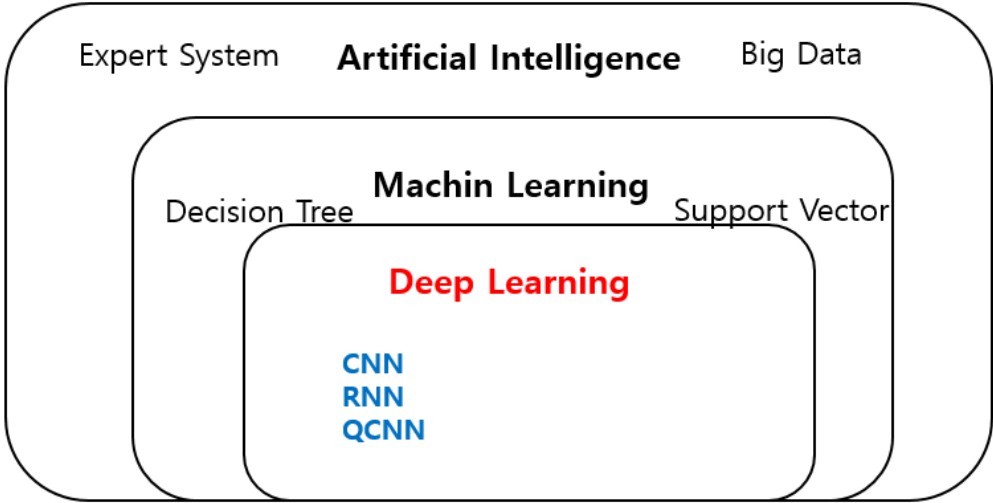

**Figure 3.** Deep learning architecture.

*2.3. Perceptron Algorithm*

As one of the artificial neural network models in artificial intelligence, the perceptron is a very old algorithm first designed by a man named Rosenblatt in 1957. In general, artificial neural network systems are modeled after the nervous systems of animals, so there are many similarities conceptually and form-wise. The perceptron of the neuron, as seen in the previous chapter, receives multiple inputs and outputs as one signal (output) [25,26]. This seems to be similar to neurons sending out electrical signals to transmit information. In the perceptron, weight plays the role of transmitting signals, such as a neuron's aquatic and volumetric bump [27]. This weight, called weight, gets assigned to each input signal, calculates it with the input signal, and outputs 1 when the total value of the signal is above the set threshold (θ; ta, ta). It is also described as the activation of the New Alliance. If it does not exceed this value, the output is 0 or −1. Each input signal is given a unique light; the larger the light, the more important the signal is. What machine learning does here is to set the value of this light (which can also be seen as a parameter). The methods vary depending on the learning algorithm, but the creation of these weights is all the same in terms of learning. Because the output value of the Perceptron is either 1 or 0 (or −1), as mentioned earlier, it can also be considered a linear classifier model. Usually, the input vectors of errors are used to calculate their linear combinations and are similar to the internal vectors covered by other posts [28].

Linear classification is similarly defined as drawing a line on a plane to separate data into classes A and B. Start with a randomly set weight first. By entering learning data into the Perceptron model, weight is improved when the classification is wrong. The meaning of improving weight is similar to solving a problem repeatedly, for instance, when we solve a math problem incorrectly until we finally get the answer right. Therefore, it is called learning. The term learning is much like the concept of learning that we actually think of. The perceptron is an appropriate algorithm when learning data can be linearly separated, because learning proceeds until all learning data are correctly classified. Linear classification means classification by line. You can see that the more you learn, the more you will change the slope of the line. As you learn, the weight continues to adjust. If you replace seta θ from the perceptron formula with -b and turn it to the left, the following formula will be derived:

$$b + w1 \times 1 + w2 \times 2 < 0 => 0$$

$$b + w1 \times 1 + w2 \times 2 > = 0 => 1$$

where b can be called bias. In the field of machine learning, it is important to prevent models from overfitting the learning data. Overfitting is where the model is so flexible that learning data is

categorized extremely well, but it does not perform properly when other data are put in. It is important to create a model that generally fits well with any data put in (see Figure 4).

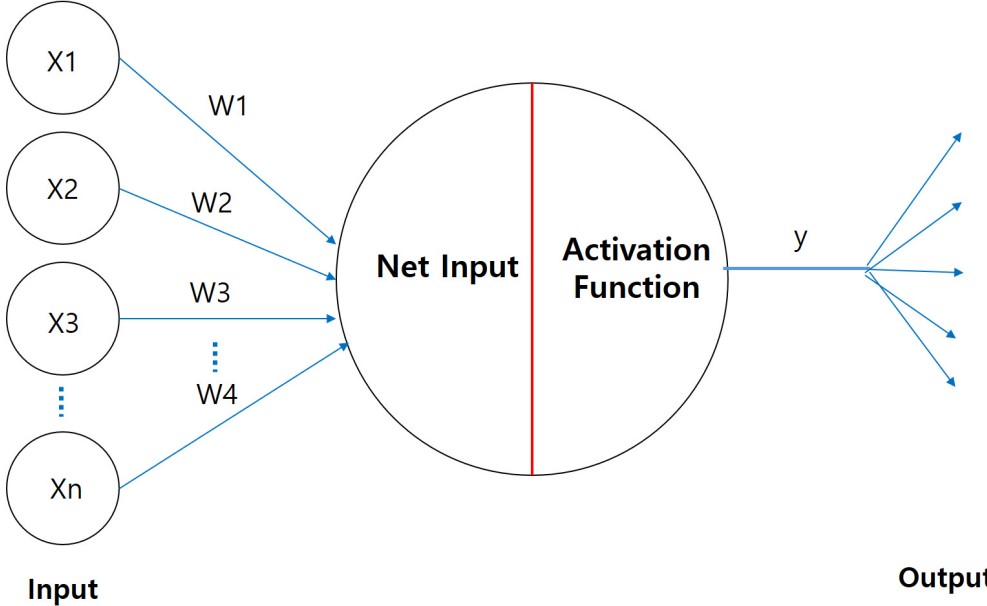

**Figure 4.** Perceptron algorithm.

*2.4. Blockchain*

The concept of blockchain was explained by a 2008 paper called Satoshi Nakamoto [6]. Blockchains are tamper-proof data technologies based on distributed computing technology. It is a technology that stores data under management in a distributed data storage environment called a "block", where small data are connected in chain form based on the P2P method, so that no one can modify it arbitrarily and anyone can view the results of changes. These blocks record all transactions that had been disseminated to users before the block was discovered and which are sent to all users in the same P2P manner and consequently cannot be modified or omitted arbitrarily. Blocks have a link between the date they were found and the previous block, and a set of these blocks is called a blockchain [29]. Simply put, it is a technology that puts together countless records in a bundle. Unlike keeping transaction records on a central server, as in the case of previously traded e-money, blockchain displays transaction records to all users and prevents forgery by comparing them with each other. You can see that Bitcoin first demonstrated the concept of blockchain, and Ethereum first implemented the concept of a smart currency. There is a close relationship between blockchain and cryptography. Blockchain is not a technology that can only be used for encryption. Encrypted money is subordinate to blockchain. Thus, technologies or services already applied with blockchain are being developed. Satoshi Nakamoto did not develop the blockchain first and then apply it to Bitcoin, but rather solved the problem of developing Bitcoin, an electronic currency system operated only by P2P, by developing and applying the blockchain [30].

This can be seen from "Bitcoin: A Peer-to-Peer Electronic Cash System," a white paper released when Bitcoin was unveiled to the world. More than 51% of nodes participating in the blockchain, or a majority of nodes, operate blockchain transactions at the same time. The theory is that, if there are 100 nodes and there is a group that controls 51 nodes, it can be controlled at will. In mining, 51% can monopolize 100% of the mining volume by ignoring 49% of the remaining blocks and insisting on the blocks they found. Blockchain technology as a whole is a key means for ensuring the reliability of the data by the "unreliable majority" of the blockchain technology users; thus, if 51% insist that they are right, they can manipulate any new blocks as well as the same ones.

In addition to the number of nodes, 51% of the miners can be manipulated, so it is important to be careful of the initial blockchain as well as the small number of miners [31]. Mature blockchains cannot be repaired, and it is only a matter of time before the blockchain's control is passed on once hacking at the level of political maneuvering is carried out. Blockchain can also be seen as an agreement convergence algorithm, ensuring that data from books distributed across each node is always present among large nodes. This ability enables nodes to run anonymously, to have poor connections, or even to involve unreliable operators (see Figure 5).

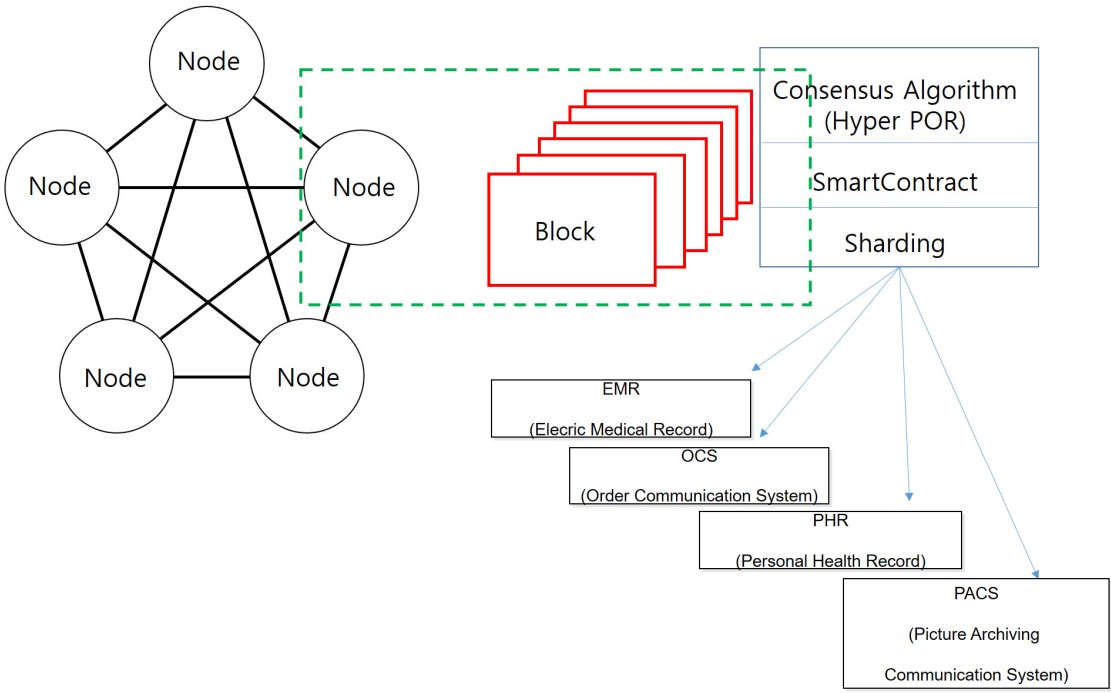

**Figure 5.** Blockchain algorithm for healthcare.

Blockchain has many algorithms. In particular, we want to apply the health information blockchain using the Hyper POR algorithm, which is an agreement algorithm among public blockchains. The reason for using the Hyper POR algorithm is that these medical data will become a PHR (Personal Health Record) in the future, requiring a large number of TPS (Translations Per Second) if many data are used. This is because many TPS performance data are required to authenticate these blockchains. Therefore, the health information blockchain needs a high-performance consensus algorithm. It also includes a smart contract function that automatically contracts. In this paper, a number of blockchain agreement algorithms, shading technologies and smart contract technologies are used to verify these health information data.

*2.5. Smart Healthcare*

Smart healthcare is an area that combines healthcare and the Internet of Things, cloud computing, big data, and artificial intelligence as the core information communication technology of the Fourth Industrial Revolution. Looking at the basic industrial structure, data generated by professional institutions such as daily life and medical institutions are collected and analyzed by data experts, and medical and healthcare companies are used again to advise and treat consumers. In the wake of the Fourth Industrial Revolution, the boundaries between previously seemingly different areas are becoming blurred. In keeping with this trend, a new area of smart healthcare has been created. Smart healthcare is rapidly emerging, as the medical service trend has changed from therapeutic purpose to disease prevention. With the rapid and widespread application of core technology ICBMs in

the Fourth Industrial Revolution to healthcare, many new operators are also jumping into the industry, in addition to existing operators [32].

Thus, various players in the industry are faced with changes, and measures are taken such as strengthening the necessary technologies and partnerships. The emergence of artificial-intelligence-based smart healthcare has caused the market to grow at a steep annual rate of 42 percent, starting at $800 million in 2015. The figure is expected to reach $6.6 billion by 2021. With artificial intelligence technology incorporated into the medical sector, more new services are expected to be created. Through artificial intelligence technology, future healthcare services will be able to analyze and interpret large amounts of genetic information on their own to predict when diseases will occur, or to help prevent disease by providing personalized diagnosis and lifestyle information. During care, conversations between doctors and patients can be automatically entered into the computer through voice recognition systems, computers can provide disease diagnostic information through stored medical charts and medical information big data, or the computer itself can analyze and learn from the patient's medical image images and provide the doctor with diagnostic information about diseases such as cancer. As a global leader in artificial intelligence, the US is focusing on enhancing the quality of medical care by promoting precision medical care using artificial intelligence. Europe is concentrating on combining artificial intelligence's medical information platform and gene analysis, whereas Japan is focusing on providing personal care and customized medical services through genetic analysis and robot strategies applying artificial intelligence. The future medical paradigm will be precision, prediction, prevention, and personalized healthcare. To make this a reality, a large amount of personal data is needed. In particular, healthcare-related personal data is highly sensitive information and should be sufficiently reliable and secure. To this end, the introduction of blockchain is necessary. The data recorded on the blockchain is virtually impossible to falsify or modify, which makes it almost impossible for personal information to be leaked. For this reason, blockchain is one of the most popular technologies in the recent smart healthcare market. Meanwhile, in January 2017, Watson Health, IBM's business unit, signed a two-year joint development agreement with the FDA (Food and Drug Administration) of the United States to use blockchain technology to share patient data securely for medical research and other purposes. Telemedicine is drawing attention from all over the world because it allows patient care anytime, anywhere. In particular, demand is expected to increase due to the acceleration of aging and the increase in patients with chronic diseases, and the size of the market will grow by an average of 14.7% per year. The global telemedicine market is divided into five major categories: telemonitoring, telemedicine counseling, telemedicine education, telemedicine training, and remote surgery. Among these, telemedicine counseling services constitute the largest market.

The acceleration of aging and the increase in diseases such as diabetes and Parkinson's disease suggest that growth can be expected in the telemonitoring service sector in the future. Telemedicine is now largely active in the United States. Representative companies leading the telemedicine market in the United States are McCesson, Philips Healthcare, GE Healthcare, and Cener, which are healthcare companies combining healthcare and IT. Characteristically, United Healthcare, a large private insurance company, is willing to compromise by offering incentives to doctors and related infrastructure companies participating in telemedicine. In Korea, MOD's smart care system, which can effectively monitor, care for, and deliver information to patients through the connection of patients and hospitals, is also being used at about 100 hospitals nationwide. As such, healthcare systems combine artificial intelligence and blockchain (see Table 1).

**Table 1.** Smart healthcare for hardware, software, and service.

| Items | Hardware | Software | Service |
|---|---|---|---|
| Purpose | Hardware systems such as robots for artificial intelligence research | Software technology as key to the study of artificial intelligence | Personalized models found |
| Related research | Wearable devices, parts, devices, reagents, etc. | Providing medical healthcare content, communication network platforms, medical information, exercise information, etc. | Genes, medical diagnosis services, genetic information |
| Speed Measurements | High | Middle | Slow |
| | Robot system for healthy strengthening | Personalized, integrated medical device services | Hardware and software mixed service required |
| Middleware | None | Middleware required | Needed |
| Technical technology | Blood sugar, blood pressure, ECG, activity measurement, chemical analysis, body fat analysis, medical sensors, field testing devices, band-necked implants | WebnisApp, nutrition management app, personal healthcare app | Personal health examination services, personal health records management systems, and healthcare services for the elderly |
| Artificial Intelligence Application Phase | Hardware Compute Platform Important | Software Logic Important | Need to develop a service model |
| Blockchain Application Phase | Total Level | Upper Level | Under Level |
| Effects of Consensus Algorithm | Middle Effect | High Effect | Low Effect |
| CNN Protocol Validation | Verification Important | More Important | Non-Important |

## 3. Verification Framework for Artificial Intelligence and Blockchain with Healthcare System

### 3.1. Issue Raising

This paper concerns the most active medical and biological research in the healthcare industry, which has recently become a hot topic. We also need to address various technical issues one by one for the artificial intelligence blockchain in the medical field. One of these is about the form and structure of medical data in circulation. Various medical data, such as clinical data and genetic data including imaging information, are vast in volume and highly complex. The amount of medical data is expected to grow exponentially, especially at a time when the development of wearable technology is causing the accumulation of health data produced by patients' PGHD (Patient Generate Health Data). Because the amount of data that can be stored in a block is very limited, it is practically impossible to store all of these data in the blockchain. In addition, if all information on an individual is stored in an irreversible form and stored as a blockchain, it could lead to potentially serious security problems. Therefore, as mentioned earlier, there is a need to distinguish clearly between on-chain data for the block and off-chain data for which actual data exists through index information as a necessary part of the medical law for the legal review of decommissioning as well as preservation of medical data. This means that the nature of the blockchain, which cannot be changed or discarded, can be vulnerable in terms of patient rights protection. Another important consideration in the blockchain ecosystem is the "right to move information." Information transfer right refers to the requirement for the data controller to send personal information to, or directly to, other information processing personnel. This refers to a system designed to strengthen control over information subjects' personal information by the European Union GDPR (General Data Protection Regulation) by specifying it in the GDPR and to ensure more options for information entities in the digital market through the balancing of relationships between information entities and data controllers. On the other hand, it is imperative that the blockchain be successfully applied to the healthcare sector since it concerns the standardization of medical data.

There are currently active studies on standardization in the field of medical information; since few medical institutions in Korea have fully applied them, however, it is necessary to watch carefully to see if current blockchain technology can fully reflect them. In particular, sufficient research on how to standardize and distribute genetic data, which are still in the developmental stage, should be supported at a time when genetic data is expanding into clinical areas. Currently, HL7's CDA (Clinical Document Architecture) is a key feature for clinical data distribution in the field of medical information. This could enable the sharing and exchange of medical information among medical institutions regardless of the type of medical institution. Meanwhile, some argue that CDM (Common Data Model) should be used as a standard format for storing data for distribution in the blockchain. For the purpose of medical research, it seems considerable time and effort are needed to have it spread to the primary hospital, since the original data can be distorted while medical institution data is standardized for medical research purposes; only data described in CDM can be distributed, and the third hospital can currently distribute. Since the use of blockchain in the healthcare sector is not solely for medical research, in-depth research is needed on the optimal alternative for data distribution. In particular, this research needs to be conducted because verification of as much information data as possible is required for applying artificial intelligence blockchain technology to verify the EMR system and PACS system accurately among medical data.

*3.2. Research Methodology*

This paper aims to verify the personal medical data of PHR, a medical practitioner in today's healthcare industry. Research is done to predict and verify medical data using artificial intelligence, deep learning, perceptron, and blockchain as necessary future medical systems. The key to future medical care is the realization of data-based customized and predictive medical care. For this to be possible, an open ecosystem that enables individuals to view, manage, and distribute their medical data anytime, anywhere must be created. Likewise, medical data address highly sensitive personal information by nature, thereby requiring significant levels of reliability and security. Achieving two goals—openness and safeness—from the perspective of data is a very difficult problem. Blockchains have recently gained much attention in the medical community as a technology for addressing the ambivalence of these data. Blockchain refers to an information and communication infrastructure wherein transaction information is distributed, recorded, and jointly managed by all users over a peer-to-peer network, unlike the traditional way of storing data on a particular central server. When blockchain technology is used in the medical sector, health information management capabilities, insurance claims, and screening processes are expected to be enhanced, including medical devices and drug distribution channel tracking; clinical trials will be more secure, clinical data sharing and utilization including personal health and health information will be enhanced, and medical information integrity and accountability tracking will be improved. Nevertheless, worldwide research on blockchain in medicine and biology is still in its infancy. In the future, specific service models for the medical sector blockchain, the types and structures of medical data in circulation, security, data levels, and standardization to be stored in the block, and a variety of studies on legal issues are needed. Thus, in the future, a national strategic roadmap for blockchain-based healthcare services should be established, for example, for the development of blockchain-based core platform technologies and governance, and for blockchain-related healthcare plans and policy development. With blockchain technology emerging in the healthcare sector, in 2016, the ONC-HIT (Office of the National Coordinator for Health Information Technology) formed an organization on the potential use of blockchain technology in the health and healthcare sector and launched research on technology and policy components. Through this, various applications regarding the interoperability of medical information and use cases through blockchain were proposed, including technical solutions to address problems in the ecosystem of medical data when blockchain is utilized in the healthcare sector. Through this study, blockchain is expected to increase health information management capabilities, streamline insurance claims and screening processes, track medical devices and drug distribution channels, improve the safety of clinical trials, increase the sharing and utilization

of research data, enhance the protection of personal health and health information, secure medical information integrity, and enhance accountability. These predictions are plausible, since the healthcare sector is, after all, a data-based industry: regarded as an area where healthcare and life sciences can be best applied to blockchain applications than any other area. In addition, increased interoperability among healthcare data operators not only improves care efficiency but also reduces time and cost through improved insurance claims processes. Furthermore, the empirical care environment is expected to be transformed into an evidence-oriented medical environment and to contribute to the realization of precision medicine. Furthermore, by returning medical data to individuals, a truly patient-oriented medical service environment is expected to be created. Various service models to be applied in the medical field are currently being tested, including medical data distribution, artificial intelligence, drug use, electronic health record development, wellness, medical payment, biomedical development, dental treatment, medical research, personal health records, genetic analysis, telemedicine, data analysis, virtual reality, psychological counseling, insurance, plastic surgery, food, and clinical testing. The examples of the development of major service models wherein blockchain technology is applied in domestic and international medical care. As such, this study proposed a research methodology that utilizes the algorithms of neural networks and verifies PHR data using a blockchain.

*3.3. Neural Network-Based Blockchain Framework*

A perceptron is treated as a set of neurons, and neural networks, with each neuron being a brain would be a sub-component of the perceptron. Just like a single-story perceptron, a neural network is like a single tree, and it is like a forest. It depends on how you define the word perceptron. In general, single-layer perceptrons are models that use the step function, a function that makes the output 1 when the threshold is exceeded, as an activation function. A multi-layer perceptron is a network with multiple layers and which uses the sigmoid function as an enable function. Neural networks can be represented from the left by input, hidden, and output layers, as shown below. Unlike both input and output layers, the hidden layer is invisible to our eyes, like a black box that is invisible internally. Because it cannot be known, it is called "hidden." It does not look much different from the perceptron discussed in the previous section.

(1)  Stair Function Step Function

The perceptron uses the step function as an activation function. This is a function that is activated when a specific threshold is exceeded. The lower left (a) is a step function and can be stopped at zero, changing to 1 at some point. You must have heard a lot about learning. These blockchain frameworks are also included.

(2)  Sigmoid Function Sigmoid Formation

The activation function used primarily by the neural network is a sigmoid function. The sigmoid function e is error, with a value of 2.7192 as a natural constant. The sigmoid function is also just a "function" for activation. It is only a transducer, which receives input and returns the output. In the neural network, only the input signal is received, converted, and transmitted. It is nonlinear in the form of a gentle curve compared to the stair function. Unlike a step function, where the output changes drastically based on a specific boundary, the sigmoid function changes smoothly, which is important in neural network learning, and uses the sigmoid function as an activation function.

(3)  Nonlinear Function

There is no benefit in using a hidden layer when using a linear function. In other words, even if the linear function is composed of multiple layers, it is equivalent to only three consecutive repetitions of the linear function. If $y = anx$ is a linear function, it is the same as $y = a(a(x))$ if it is constructed in three layers, which is the same as $y = a3(x)$. This implies that it is meaningless to construct a network with linear functions without the hidden layer.

(4) Multidimensional Arrangement

The calculation of multidimensional arrangements involves using matrix operations to make the values of the weights described earlier easier to determine. One or two layers of the neural network may be calculated by human beings, but it is impossible to calculate the weights of neural networks of numerous neurons at more than one level. Matrix computation is what helps make this easy even for computing. An important concept is multiplication between parallel multidimensional arrangements. The values of b and c inside the multiplication operation must match, and the result will be in the form a x d when multiplication is performed. This applies equally when one side is in a one-dimensional array (see Figure 6).

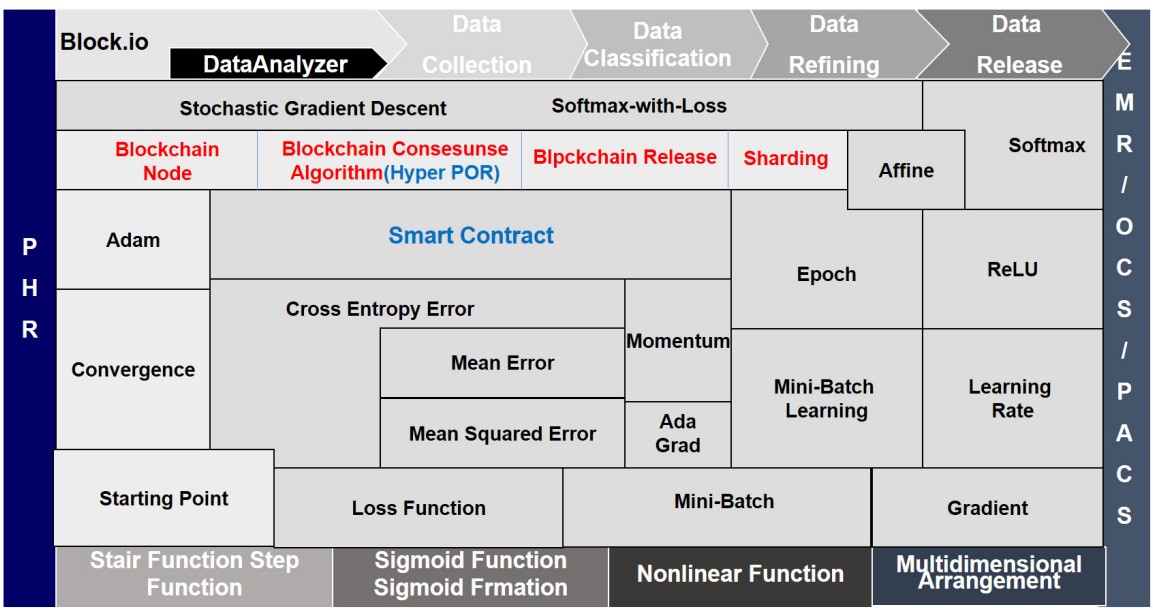

**Figure 6.** CNN (Convolutional Neural Networks) intelligent agent cloud architecture to increase healthcare data readings.

Figure 6 presents an architecture that verifies medical data using artificial intelligence algorithms in multi-dimensional arrays. This works with the Hyper POR algorithm methodology, a blockchain agreement algorithm. This data is also what creates a decentralized system with shading technology. The CNN algorithm will also validate PACS (Picture Archiving Communication System), MRI (Magnetic Resonance Imaging) and CT (Computed Tomography) data. After verifying the original data, the data is then placed in the block of the blockchain, then in smart contracts, and is distributed separately by consensus algorithms. It is an architecture for this. This paper validates these architectures and determines cross entropy error, mean error, mean squared error, and loss function. It first presents the verification data based on the neural network and then the architecture based on the blockchain agreement algorithm after verification.

### 3.3.1. Error Backpropagation Blockchain Framework

In the error inverted wave blockchain framework, the result is determined through the activation function, updating the weight from the input to the output. This concept is called a net wave, literally propagating and sending input values to the front. The output value will not be accurate even if we send the radio waves at random once. Although the weight value we randomly set has been updated by the input once, there can be many problems. The inverted wave method involves updating the weight by sending the error back in the input direction through the result value. Of course, more errors will be returned to the nodes that have more greatly affected the results. The figure above shows the resultant value in the output layer in the direction in which the input is received. The resultant value

has an error, and the inverted wave is used to reverse the error back to the hidden layer and the input layer, to apply the error that occurred in the output while calculating the weight. A single turn is called a one-epoch cycle; as you increase the epoch, the weight continues to be updated and learned, and the error gradually decreases. The resulting value from the output layer is marked with an error of 0.6. Because the previous node in the neural network delivered values of 3 and 2 to the output layer, the nodes above and below can be said to have affected 60% and 40%, respectively. We can distribute the weight equally; since it has an effect, however, we should reverse the error again. By multiplying the error, 0.6, by 0.6 and 0.4, the error is transferred as 0.36 to the node above and as 0.24 to the node below. Error inversion means literally propagating again, going back to the error in this way.

### 3.3.2. Error Backpropagation Framework

Error reverse wave blockchain framing continues to update the error by reversing it, because the error affects the adjustment of light to produce a better result value through the neural network. As in the example above, a simple neural network would be very simple in calculating the error, but an effective neural network is never that simple. If dozens or hundreds of neural nodes are connected, you have to calculate the weight of a particular node from a combination of numerous weights. An efficient method will be needed. Slope descent is a method designed to do this efficiently, because it takes a long time to calculate all the weight combinations in so many neural networks. Use of the slope descent method may not provide the correct answer. Since this is a step-by-step approach, it is a way of continuing to find answers until satisfactory accuracy is achieved. Neural network error is a method of finding the lowest error by sloping down, and it speeds up the calculation of the neural network. This involves locating the lowest point, moving slightly to the right of the $x$-axis. Nevertheless, the problem is finding this small area. If it is just a small area, if you move too much, you will pass the lowest point. On the other hand, small movements or many movements may cause the lowest point to be found. The other problem is that you can find the wrong lowest point. If there are multiple dimensions, it will not be a simple secondary function, as seen above. This method is designed to differentiate the error function from its current position in order to find the direction of the code (see Figure 7).

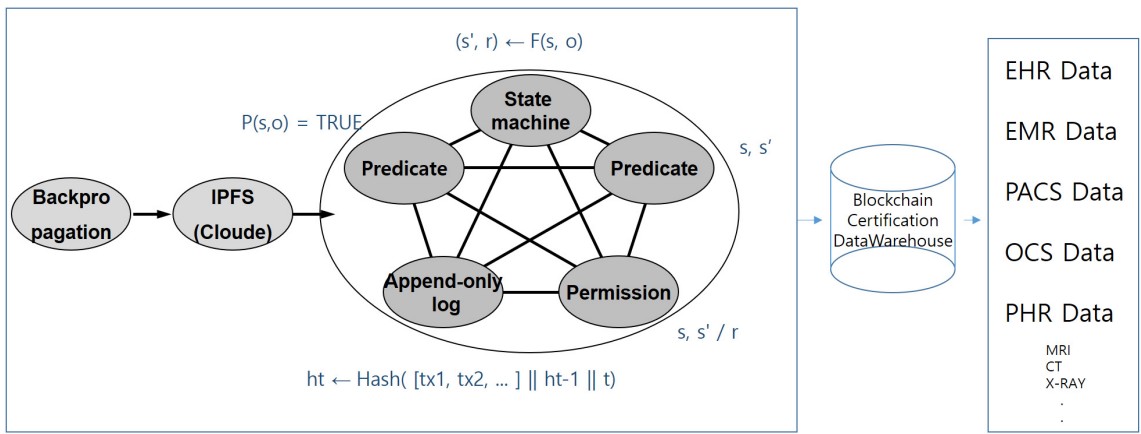

**Figure 7.** Error backpropagation framework for healthcare.

In addition, backpropagation placement is used to draw data architectures based on IPFS (Inter Planetary File System) data. Artificial intelligence data is also loaded into the cloud system to verify numerous medical data. In this paper, a blockchain verification data warehouse is created, and these data are mounted on EHR, EMR, PACS, OCE data etc. All medical data are also mounted, based on MRI, CT and X-ray data. It is very important to verify the convergence of these data initially, because it can be very harmful for inconsistent data to be stored in blockchain blocks. These data-based blockchains are loaded.

Machine learning is also about finding the most predictable model, as mentioned earlier. In deep learning, the value of the loss function is obtained from the set of training data, because the value of the loss function for the weighting parameter must be found at the lowest value. The values obtained are then used to determine the performance of the model with one indicator. What is obtained from the preceding cross-entropy error is the loss function for one case, which, if you are learning about approximately 60,000 cases, such as with mnist data, will require 60,000 cross-entropy errors to be calculated. If you go to a real big data environment, this could be 6 million, 60 million, or 600 million data, not 60,000 data, and it would take a very long time to calculate these one by one. Therefore, the method takes some of them and uses them as a whole "near value" instead of just one calculation at a time to perform the learning. Part of it is called mini-batch. Mini-batch learning involves randomly picking some of the training data (see Figure 8).

```python
from mnist import load_mnist

(x_train, t_train), (x_test, t_test) = load_mnist(normalize=True, one_hot_lab
el=True)

print(x_train.shape)
print(t_train.shape)
(60000, 784)
(60000, 10)
train_size = x_train.shape[0]
batch_size = 10
batch_mask = np.random.choice(train_size, batch_size)

x_batch = x_train[batch_mask]
t_batch = t_train[batch_mask]
np.random.choice(train_size, batch_size)
array([24632, 13808, 21266, 43860, 34652, 15548, 58670, 47186, 30791, 22296])
```

**Figure 8.** Source result for mini-batch learning.

The reason for using the loss function is for ultimately obtaining a weighting parameter with high accuracy. This finds the smallest possible value of the loss function when obtaining the optimum weight and deflection, and then calculates the differential (slope) and repeats the process of slowly updating the value of the parameter for the weight and deflection through such a differential value. The differential of the loss function is the change in the loss function when the value of the weighting parameter is changed. If the value of the loss function is negative, the value in the positive direction should be decreased; if it is positive, the value in the negative direction should be decreased. The weight parameter should no longer be updated when the differential value is zero and there is no change. Accuracy is not appropriate, because the value of the loss function must also change in response to small changes in the parameter. This is the same reason the stair function has a large value of zero and smoothly changes it into an activation function, such as a sigmoid function (see Figure 9).

```python
def cross_entropy_error(y,t):
    if y.ndim == 1:
        t = t.reshpae(1, t.size)
        y = y.reshape(1, y.size)

    batch_size = y.shape[0]
    return -np.sum(np.log(y[np.arange(batch_size),t]))/batch_size
```

**Figure 9.** Source result for entropy.

To find the optimal weight and bias, learning is carried out, with the loss function having the highest value. The larger the dimension of the weight, the harder it becomes to find the point at which this loss function has the highest value. The slope can be used to find where the peak value of the function is located; this is called the slope descent method. The slope indicates whether the peak value of the loss function is there, including the minimum direction. At the point of peak value, the slope is zero. Note that the slope can be zero, and this point is not necessarily the highest. This is also a problem that can arise from the slope descent method (see Figure 10). The text even studied the source codes for various frameworks, presenting a framework model.

```python
def numerical_gradient(f, x):
    h = 1e-4
    grad = np.zeros_like(x)

    for idx in range(x.size):
        tmp_val = x[idx]

        #f(x+h)
        x[idx] = tmp_val + h
        fxh1 = f(x)

        #f(x-h)
        x[idx] = tmp_val - h
        fxh2 = f(x)

        grad[idx] = (fxh1 - fxh2) / (2*h)
        x[idx] = tmp_val
```

**Figure 10.** Source result for gradient.

This source code is an example of the actual machine learning code. It is difficult to disclose all sources in this paper. However, it was used as data to extract some source codes and verify them. These source codes receive variables as mnist imports load_mnost. The program used Python software. The source code was verified by receiving a random choice. This contains procedures for verifying the necessary sources.

*3.4. CNN Intelligent Agent Cloud Architecture Flowchart*

This section of the paper verifies a CNN of blockchain-based artificial intelligence.

Step (1) The CNN intelligent agent cloud architecture goes to Block.io. This is for data analysis. Data is collected and classified by type. This also refines data. Distribute final data.

Step (2) Take the blockchain algorithm that verifies these refined data. The blockchain uses Hyper POR for the agreement algorithm. The Hyper POR algorithm works by verifying the business partner. Then set the generation block that verifies in the middle. After that, add shading technology to do distributed computing.

Step (3) Verified data can be verified by a CNN verification algorithm using this refined blockchain technique. In particular, EMR and PACS data are verified by RELU, learning rate and epoch verification algorithms.

Step (4) This is a step in verifying the collected data. Verification is done using the stair function step function, sigmoid function sigmoid fraction, nonlinear function and multi dimension arrangement functions.

Step (5) Finally, Step 5 verifies verified data with probabilities. It verifies the convergence of the collected data and ultimately delivers the most error-prone EMR, PACS, and so on (see Figure 11).

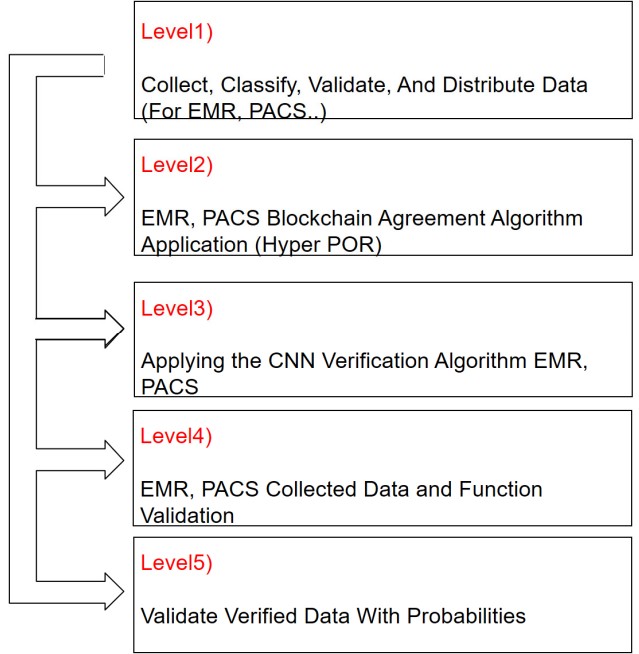

**Figure 11.** CNN intelligent agent cloud architecture flowchart.

*3.5. UML Diagram, A Neuron Network Blockchain for PHR Verification (Sequence Diagram)*

The sequence diagram consists of a time series. In other words, it shows object interactions arranged in chronological order.

Express objects and classes that accompany a scenario and a series of messages exchanged between objects are essential for performing scenario functions. Sequence diagrams generally relate to the realization of the use case of logical views of the system under development. Also called event diagrams or event scenarios, sequence diagrams are a little more readable than the class diagram. If a class diagram describes how it is actually organized internally, then the sequence diagram shows how it works. It is something that does not change over time, and this is a static relationship. Class Diagram A has a dynamic relationship or a sequence diagram that varies with time. The sequence diagram moves from PHR data to BlockIO objects. Then, the neural network shows the sequence diagram of sending and receiving through put, remote, and send messages (see Figure 12).

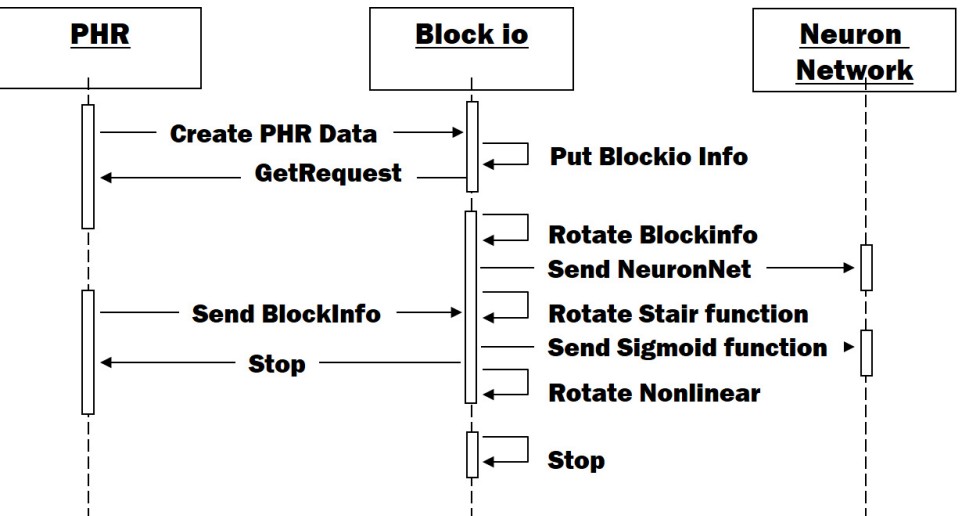

**Figure 12.** UML diagram, a neural network blockchain for phr verification (sequence diagram).

*3.6. Neuron Network Blockchain and General PHR System Performance Detection*

Performance testing was performed by applying blockchain architecture with a neural network. This performance test was conducted in a variety of environments. Performance evaluation was conducted by combining the artificial intelligence neural network and the existing blockchain performance evaluation. Several of the concepts that make up the artificial intelligence blockchain are directly related to speed.

1. Transactions per second (TPS)
2. Block generation time
3. Confirmed time

The TPS is an artificial intelligence blockchain; it is the speed at which an employee's business receives and processes a transaction head. In some cases, it is possible to process more ledgers during different business hours between different members. The processed head is put in a box and delivered to another employee. The block creation time is also the time it takes for another employee to seal the box and affix the approval seal. It takes time to receive payment, so you can think of it as a constant time per box. The fixed time is the time to keep stamping along the approval line. The company is actually horizontal, so anyone can affix the approval stamp; the more stamps affixed, the less likely the box will become invalid, and the greater the convergence on 0. Transactions per second is the number of transactions per second. Most widely known as the speed indicator for blockchain, the TPS tells us how many transactions can be processed and stored per second. TPS may depend on the design of blockchain software as well as the hardware performance and network performance that drives the blockchain and creates/verifies blocks. It may also vary depending on the type of transaction. For example, a smart currency transaction requires more operations for a simple remittance, so it can only take longer. A currency that performs the same function may require more operations depending on the actual code implementation and may possibly take longer. There can be two types of TPS. The maximum TPS shows how many transactions are processed and stored per second when overloaded. Therefore, it is a very good indicator for evaluating speed. The average TPS was also evaluated for performance. To reduce the existing evaluation time, PHR data without a security system and PHR data with an artificial intelligence blockchain were compared with performance tests.

In addition, 100 nodes were configured, and one performance test was performed to verify the existing blockchain's TPS. Based on the structure of the IPFS (Inter Planetary File System), an average performance of 2400 TPS was shown (see Figure 13). Network performance was also determined using the blockchain neuron network.

In addition, we applied the neural network blockchain's consensus algorithm to verify the performance process for about 50 times. When this is done, the average TPS is 3000 TPS, showing a 600-TPS rise (see Figure 14).

Finally, when the neural network blockchain is applied, 100 nodes are configured for the performance test. At this point, you can see almost the same 3000 TPS performance data (see Figure 15).

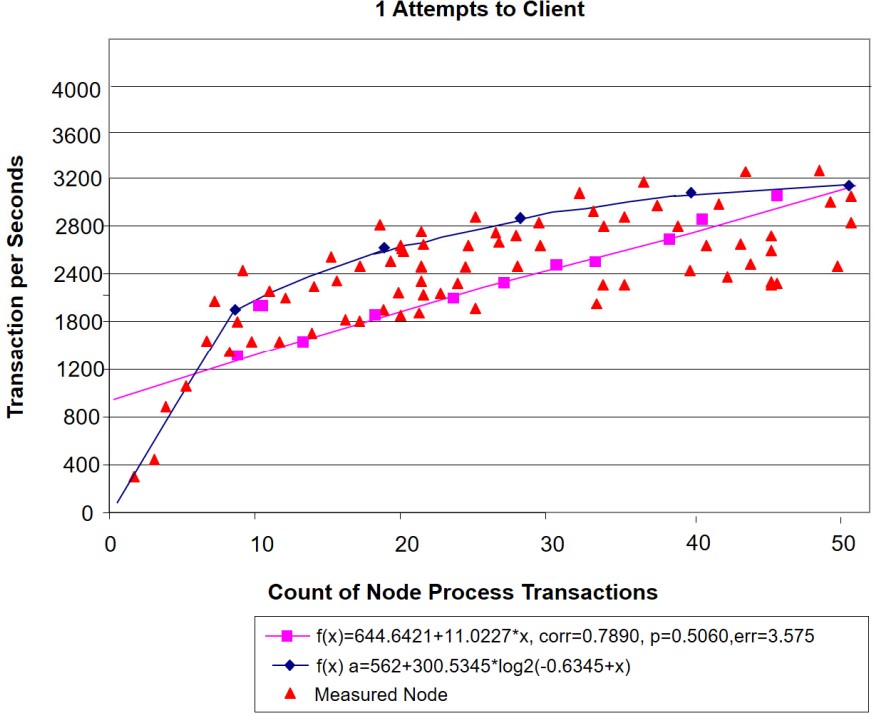

**Figure 13.** Existing security system application model (one round).

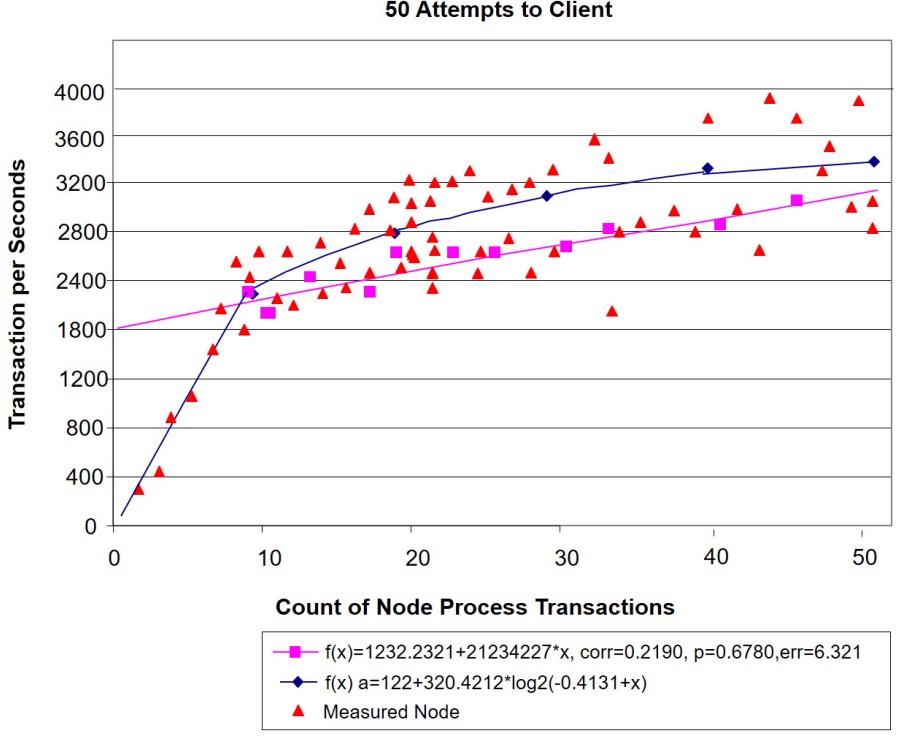

**Figure 14.** Existing security system application model (50 rounds).

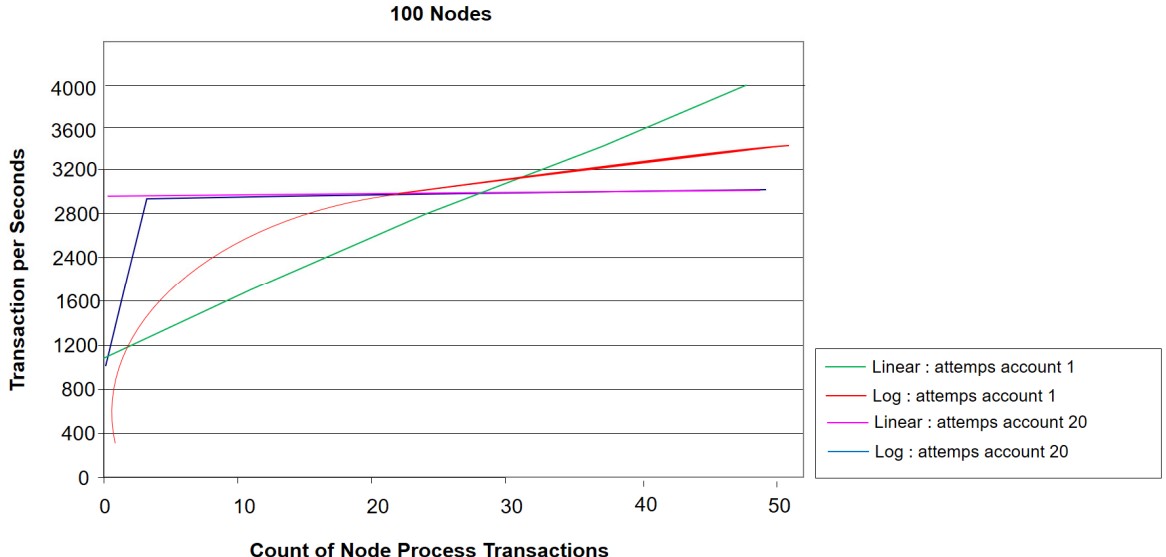

**Figure 15.** Average value for 100-node.

### 3.7. Implementing the Neuron Network Blockchain dApp

The blockchain dApp is applied to the neural network. As the development environment, DB (DataBase) used AWS (Amazon Web Services) cloud systems, and we built up to 100 nodes. The first screen shows the personal information record. Most hospitals and medical institutions contain personal names, mobile numbers, blood types, and vision and disease information. These individuals' health information sometimes includes sensitive information. Thus, personal information should be dealt with in a more in-depth manner. Personal information is becoming increasingly important for management by GDPR, etc. in Europe. Such personal information is managed through encryption and masking processing of critical information. Figure 16 shows the screen containing such personal information.

The screen in Figure 17 shows that unstructured and video data, which are PACS information from actual individuals, are classified as personally identifiable data. The application of unstructured data such as video to these blockchain blocks causes the degradation of server performance. Therefore, the video data is stored and applied in IPFS files according to the address value. This allows for quick registration and recall of address values only. As such, it shows fast performance figures.

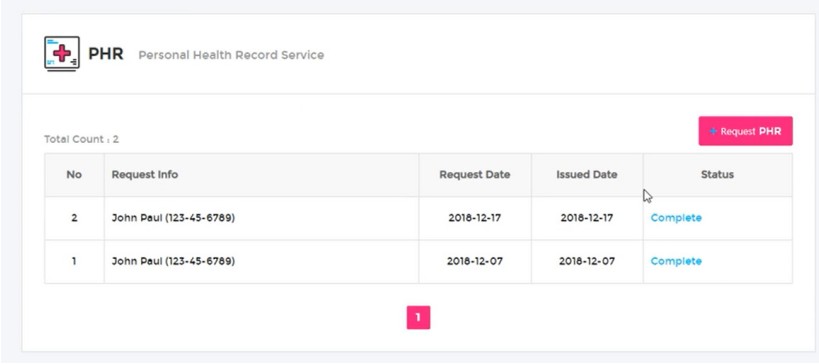

(**a**) First request

**Figure 16.** *Cont*.

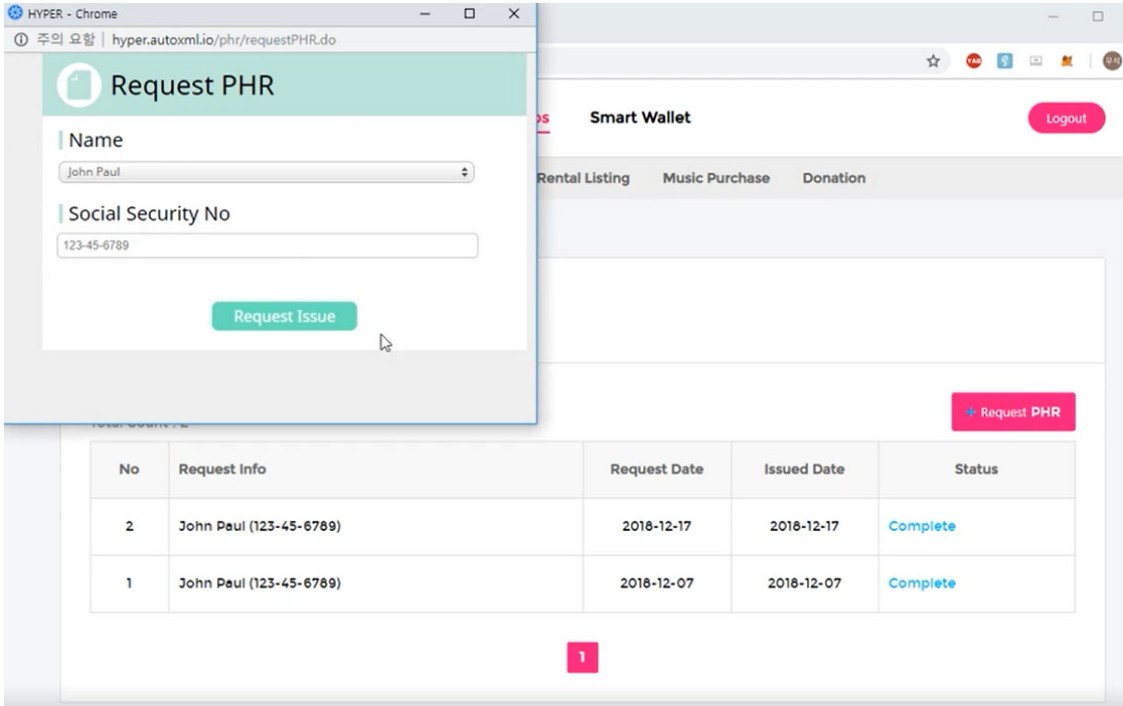

(**b**) Request PHR

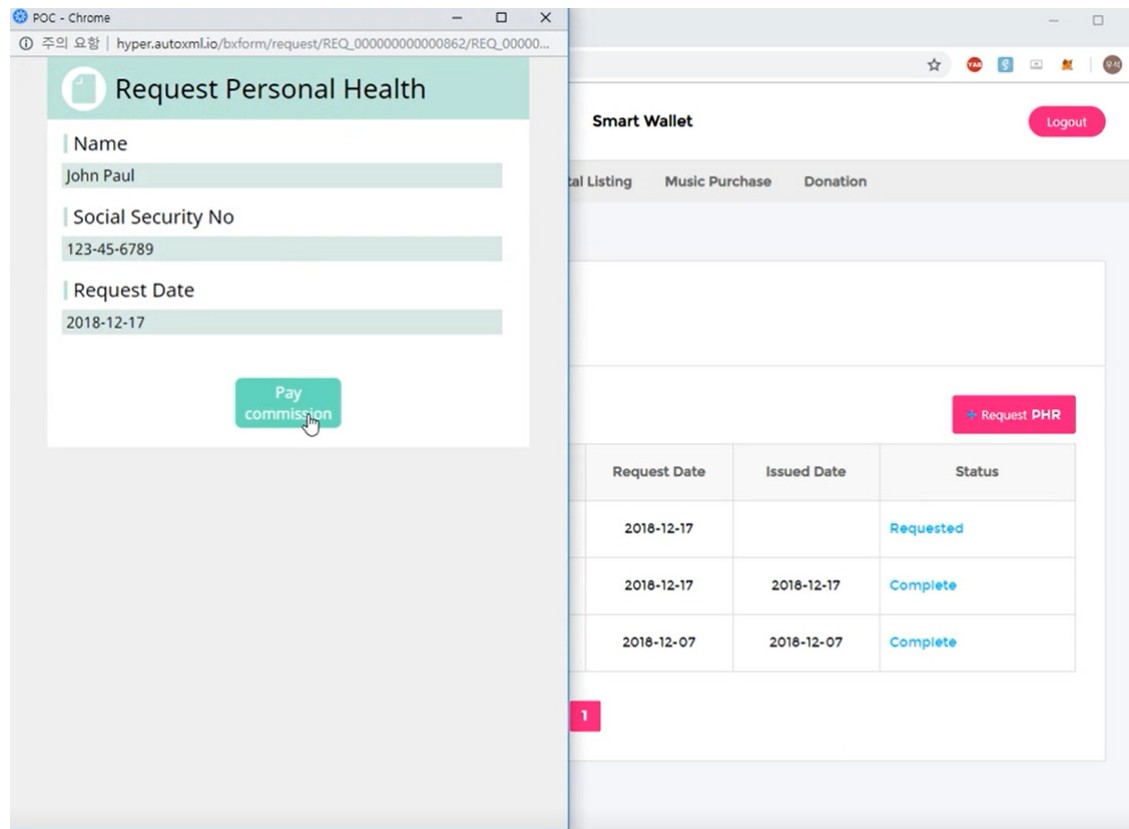

(**c**) Request personal health

**Figure 16.** *Cont.*

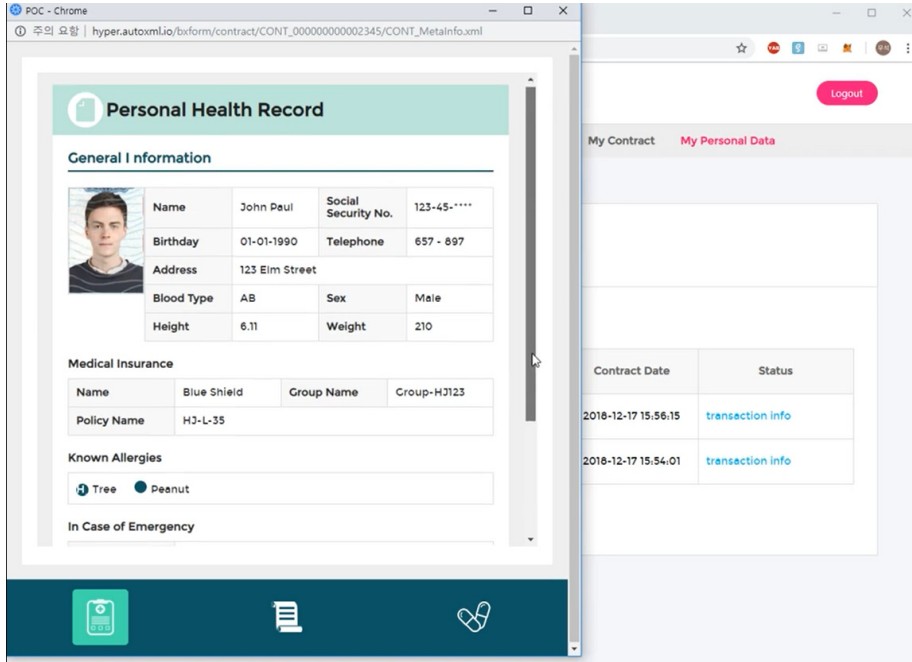

(**d**) PHR privacy screen

**Figure 16.** Blockchain PHR privacy screen using the neural network.

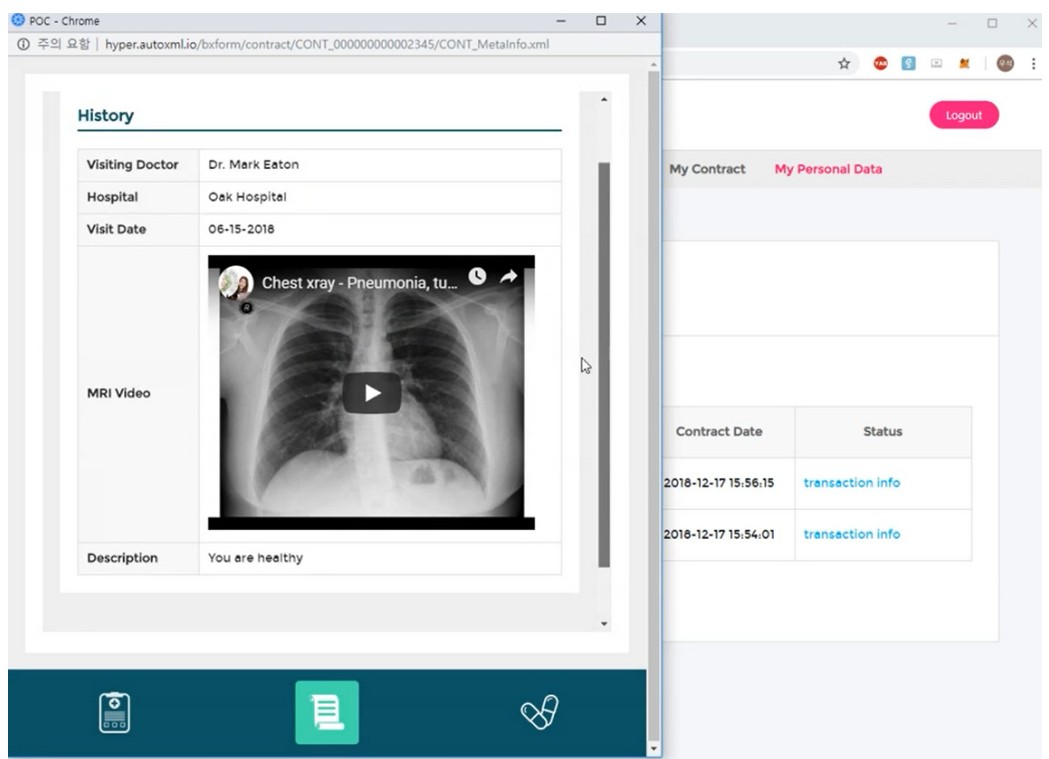

(**a**) PHR privacy history

**Figure 17.** *Cont.*

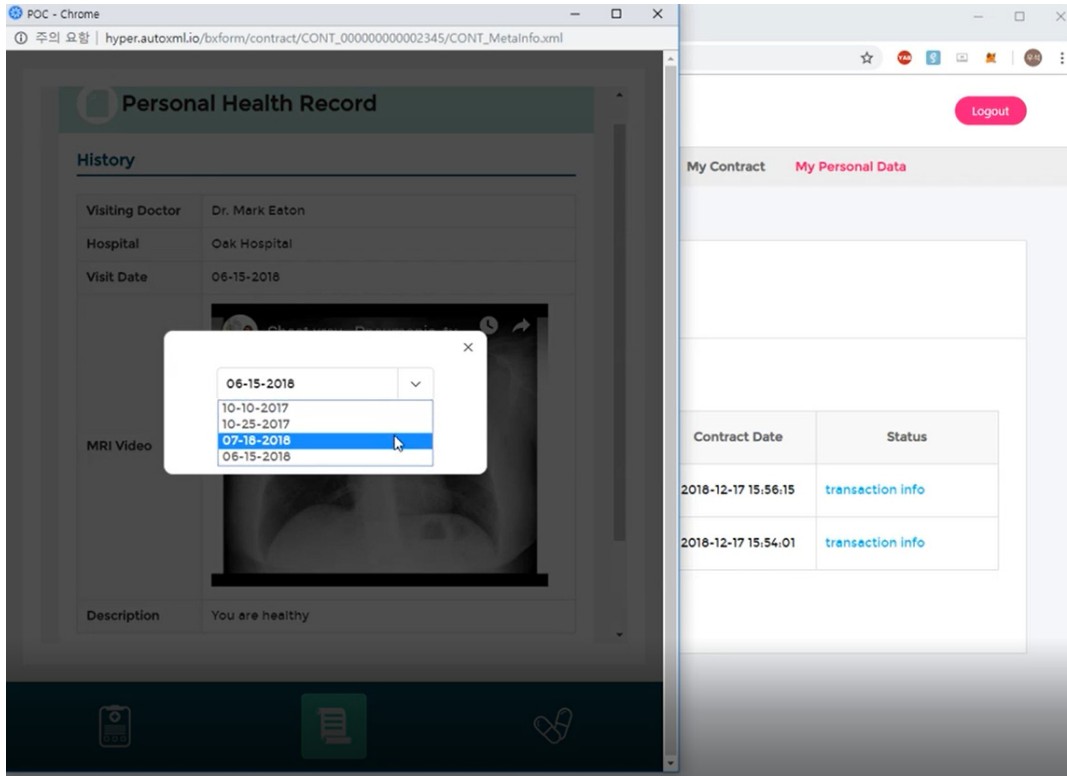

(**b**) PHR privacy day

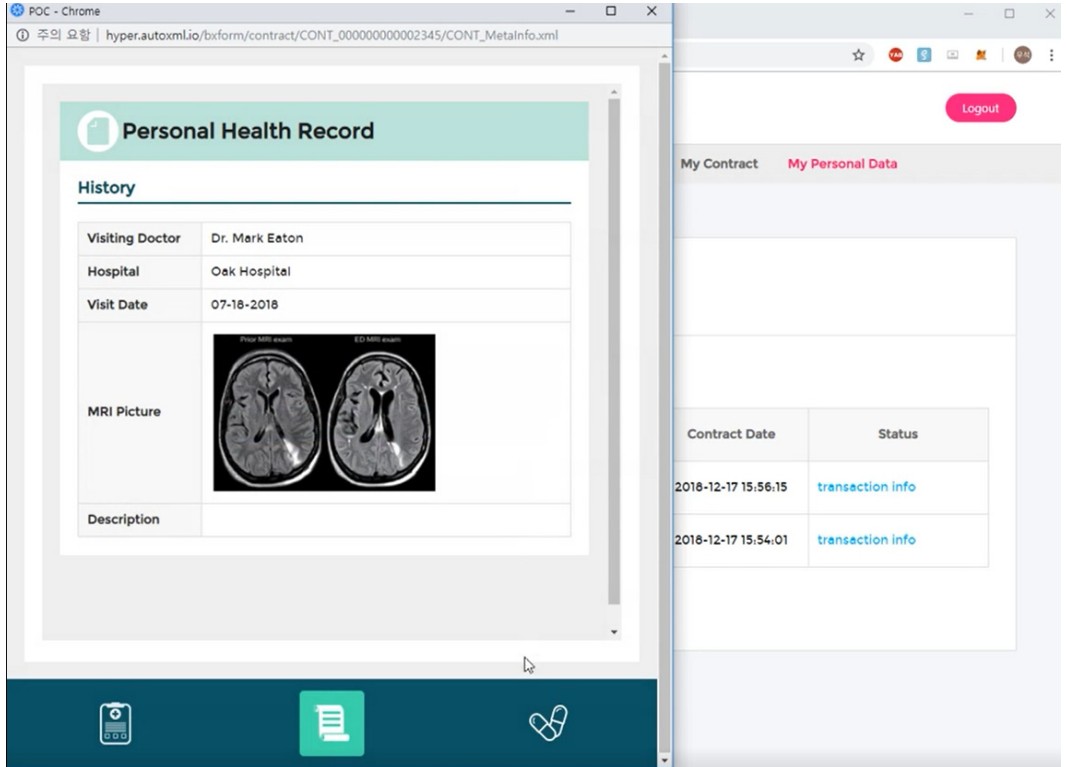

(**c**) Personal health record: history

**Figure 17.** *Cont.*

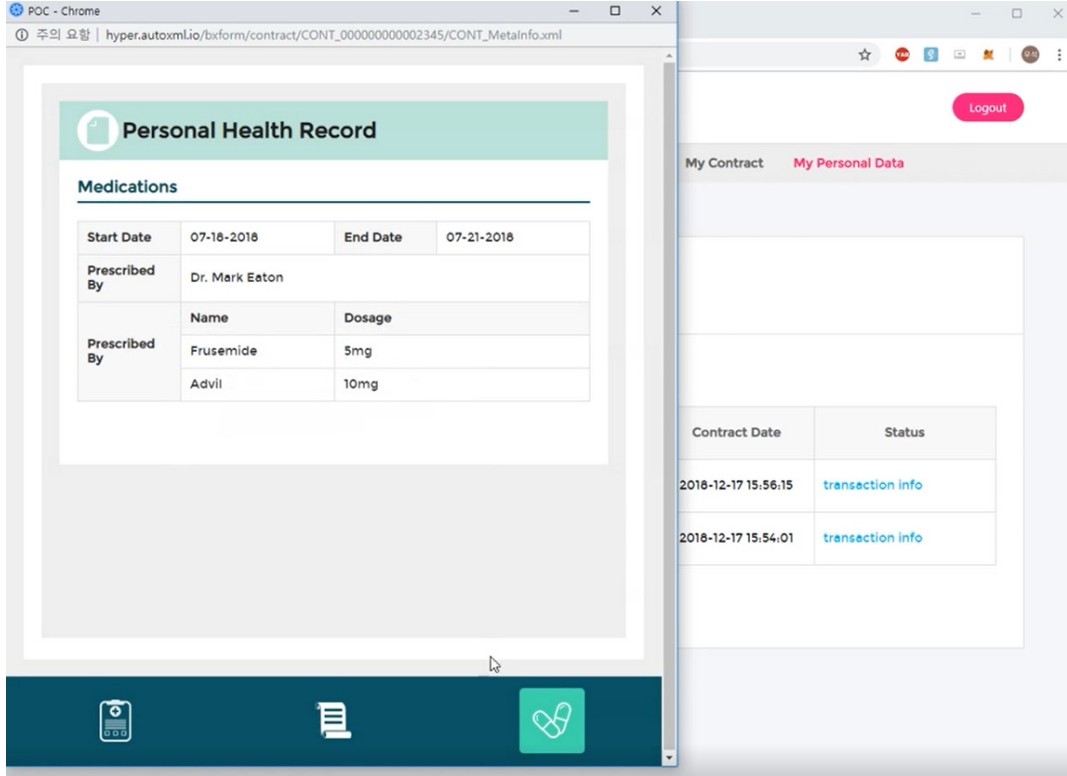

(**d**) Personal health record: medications

**Figure 17.** Blockchain PHR privacy and X-ray screen using the neuron network.

The last file also shows the smart country content. By signing a reliable contract through an electronic contract, remote care without hospital intervention has also been made possible. The initial value is the only one stored as BlockNum for the first time. A value such as a similar ID is stored. You can also see an electronic signature with the hash value. The hash algorithm uses SHA-512, and it is actually communicating with AWS's work site server. The transaction is explained through the final collection information (see Figure 18). Also, Video S1 shows Blockchain for healthcare system (focusing on the Personal Health Records).

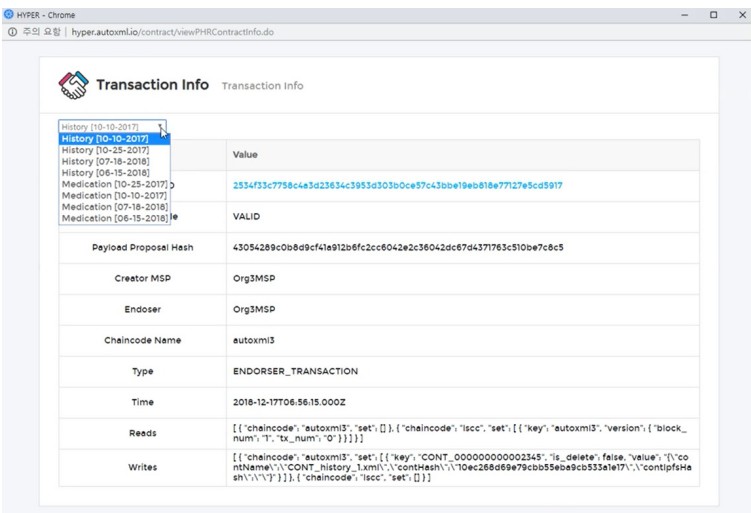

(**a**) PHR info and contract screen

**Figure 18.** *Cont.*

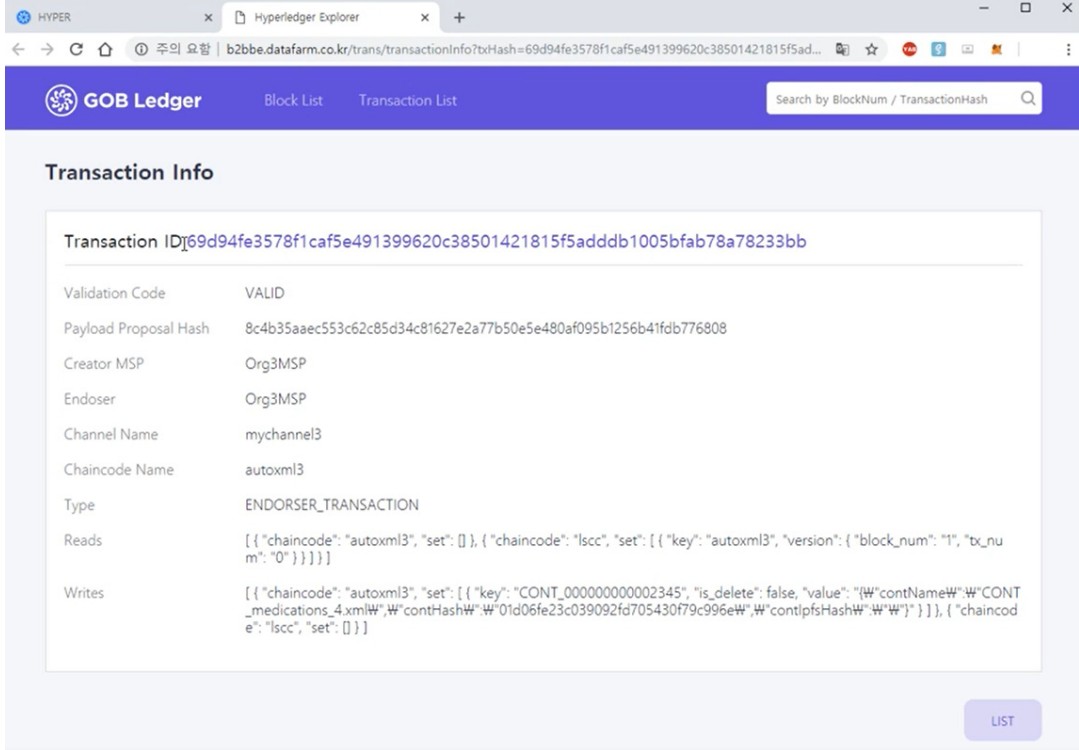

(**b**) PHR smart contract screen

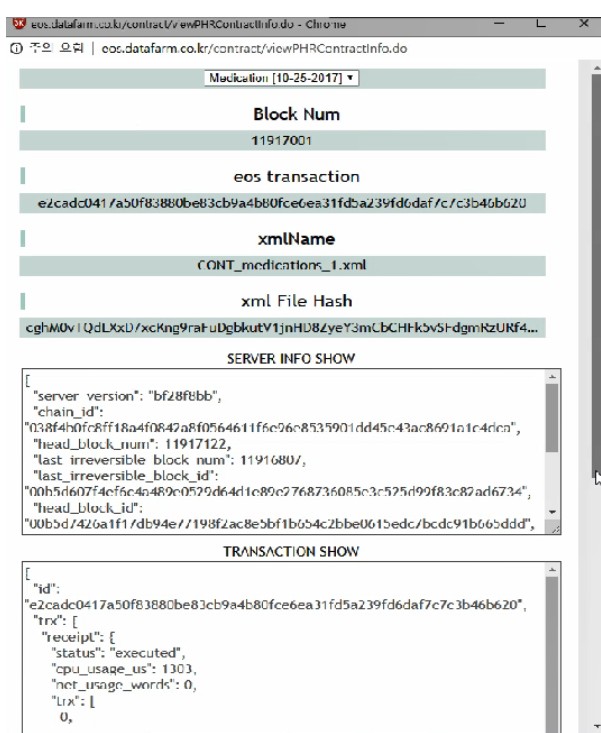

(**c**) Contract screen using the neuron network

**Figure 18.** Blockchain PHR smart contract screen using the neural network.

## 4. Conclusions and Future Work

Artificial intelligence blockchain technology is not something that should be applied to everything. Medical information systems can be verified, especially by using blockchain, and services that are

personalized using artificial intelligence services can be used in real time. This study showed how to verify it using accurate image data extraction and a blockchain based on PHR data from healthcare systems by using a wide range of etched artificial intelligence and blockchain technologies. It also presented various artificial intelligence blockchain frameworks. Performance verification is shown as well, using the Error Backpropagation Blockchain Framework and Error Backpropagation Framework. Since no one is sure that the values of de-centralization, transparency, and so on pursued by blockchain technology are absolutely correct, unconditional trust in blockchain technology should be avoided. In fact, blockchain technology is still immature in terms of security and scalability, which act as a threshold for the application of blockchain technology in the medical industry.

Moreover, this study overcame these uncertainties. In general, the capacity of medical information imaging needs to be very large, so unless the issue of the scalability of blockchain technology is resolved, there is a difficulty in applying it immediately. The technology also guarantees integrity, such as prevention of forgery, but the security of individual blocks is not much different from that of traditional cryptography; thus, it ensures higher security than any other technology, since it uses artificial intelligence blockchain technology to store medical information in a block and link its users. In addition, blockchain technology has limitations, since it is difficult to respond flexibly to changes in the legal system for medical information or personal information that changes from time to time. Despite the obvious limitations, data stored on the blockchain using artificial intelligence blockchain is non-reversible. This is because it is impossible to delete, including falsification. In particular, the issue of personal health information, which runs counter to the recently implemented GDPR, has been resolved by blockchain technology in the medical industry to work properly. Nevertheless, blockchain technology revolutionizes the medical industry in a variety of ways. The reason is that the problems facing the current medical industry as described earlier are chronic ills and have not been solved by other technologies so far. There is clearly a role that blockchain technology can play in the medical industry in this situation, wherein hospital-oriented medical information is shifted to patients; problems regarding insurance claims, the review process, and the drug distribution process are continuously raised; and medical information is increasing exponentially due to the development of ICT technologies such as IoT. We do not know when, but we look forward to seeing blockchain truly revolutionizing the medical industry by supplementing its technological and institutional limitations.

Thus, this study went through the method of verifying the perceptron theory and used experimentation to confirm the blockchain consensus algorithm and artificial intelligence neuron network, among various technologies of the Fourth Industrial Revolution. Future artificial intelligence, big data, and blockchains are not considered to be different. This is because each technology is unique, yet they work complementarily. The introduction of blockchain technology does not only provide convenience to the industry. There are certainly skeptical opinions about the introduction of artificial intelligence and blockchain, as there are some negative aspects of the technology. First, there is the question of whether security and reliability, which are the typical advantages of blockchain, are efficient when converted into costs. It takes considerable effort for the miners in the network to release encrypted information to verify the information contained in the block. The biggest cost for the miner is the electricity bill, as the decryption requires a lot of electrical energy to run and cool the computer. Because there is only one participant who makes a block and gets paid for it, the remaining participants, except one, are wasting energy. The safety of information can also be raised. Blockchain safety has also been ensured due to the network structure, which requires the hacking of more than half of the peers in the network to enable hacking of the entire blockchain system. Japan's Mount Goggs, the world's largest bourse, was hacked in February 2014, and a total of 850,000 bitcoins disappeared; this resulted in a market loss of more than $460 million.

Although the event occurred because the infrastructure of the blockchain itself was weak, it led to the question of whether the blockchain was safer than the existing system. Likewise, Coincheck, one of Japan's largest virtual currency exchanges, was hacked on Jan. 26, 2018, resulting in losses worth approximately $530 million. The missing virtual currency from Coincheck was not as well-known as

bitcoin, but it was the 10th largest in market capitalization, affecting other virtual currencies. In addition, blockchain introduced by companies is a private consortium blockchain that limits participants; in fact, a public blockchain with a relatively large number of participants is less vulnerable to hacking. Ironically, instead of limiting participants and increasing security, they are more likely to be exposed to the risk of hacking. Another problem is the lack of legal and institutional mechanisms to support the activation of blockchain technology. Since the current privacy law focuses on a centralized management system, it will inevitably conflict with blockchain technology, which is inherent in decentralization/centralization. For example, if laws prescribe a period of data retention, it conflicts with blockchain characteristics that make it virtually impossible to delete transaction records. Since the introduction of blockchain is in its early stages, strategies should be prepared to respond quickly to blockchain development paradigms and environmental changes. Especially from a legal and institutional perspective, it is necessary to identify expected problems after the introduction of blockchain at a national level and prepare them in advance, as there are difficulties in complying with the current statutory regulations when introducing blockchain, such as the Electronic Financial Transactions Act, the Privacy Act, and the Credit Information Act. Since it is a technology in the early stages of introduction, it is important to prepare the legal and institutional basis when looking at the current situation wherein the technology gap is not significant. The introduction of blockchain has the advantage of providing new conveniences in life and industry. This technology can be evaluated positively as an engineer, because it can streamline the transaction process to reduce transaction time, reduce the arbitrage costs, and provide a new IT ecosystem environment, which can lead to innovation.

In addition, since the introduction of blockchain technology is still in its early stages, it is true that there are questions about the risks and safety of the system. Efforts should be made to develop technologies with a more secure and proven infrastructure. Just as it took a long time for the Internet to take root sufficiently in people's lives when it first appeared, a long time and effort will be needed for blockchain to be introduced into the industry in earnest, to be grafted throughout our lives, and even to affect the healthcare industry. As the actual service of the frame blockchain of artificial intelligence is expected to be highly effective and impactful, post-commercial appearance should be reconsidered to make it safer and more convenient for people to use. This research was conducted to overcome these shortcomings.

**Supplementary Materials:** The following are available online at http://www.mdpi.com/2079-9292/9/5/763/s1, Video S1: Blockchain for Healthcare System: Focusing on the Personal Health Records.

**Author Contributions:** Conceptualization, S.-K.K. and J.-H.H.; Data curation, S.-K.K.; Formal analysis, S.-K.K. and J.-H.H.; Funding acquisition, J.-H.H.; Investigation, S.-K.K.; Methodology, S.-K.K. and J.-H.H.; Project administration, J.-H.H.; Resources, S.-K.K.; Software, S.-K.K. and J.-H.H.; Supervision, J.-H.H.; Validation, J.-H.H.; Visualization, S.-K.K. and J.-H.H.; Writing—original draft, S.-K.K. and J.-H.H.; Writing—review & editing, J.-H.H. All authors have read and agreed to the published version of the manuscript.

**Funding:** This work was supported by the National Research Foundation of Korea (NRF) Grant funded by the Korean Government (MSIT) (No.2017R1C1B5077157). Also, this research was supported by the Energy Cloud R&D Program through the National Research Foundation of Korea (NRF) funded by the Ministry of Science, ICT (NRF-2019M3F2A1073385).

**Conflicts of Interest:** The authors declare no conflict of interest.

## Abbreviations

| | |
|---|---|
| PHR | Personal Health Record |
| APIs | Application Programming Interface |
| SQL | Structured Query Language |
| SLA | Service-Level Agreement |
| VMs | Virtual Machines |
| EHR | Electric Health Record |
| TPS | Transaction Per Second |
| EMR | Electric Medical Record |
| PACS | Picture Archiving Communication System |

| OCS | Order Communication System |
| MRI | Magnetic Resonance Imaging |
| CT | Computed Tomography |
| GDPR | General Data Protection Regulation |
| SVMs | Support Vector Machines |
| NBCs | Neumann Boundary Conditions |
| DLT | Distributed Ledger Technology |
| RELU | Rectified Linear Unit |
| RNNs | Recurrent Neural Networks |
| CNNs | Convolutional Neural Networks |
| DBN | Deep Belief Network |
| NCS | Collaboration Neural Network |
| FDA | Food and Drug Administration |
| PGHD | Patient Generate Health Data |
| CDA | Clinical Document Architecture |
| CDM | Common Data Model |
| ONC-HIT | Office of the National Coordinator for Health Information Technology |
| DB | DataBase |
| AWS | Amazon Web Services |

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
