# Peer review of "Artificial Neural Network Blockchain Techniques for Healthcare System: Focusing on the Personal Health Records"

_electronics, doi:10.3390/electronics9050763_

Round 1

Reviewer 1 Report

Thank you for the opportunity to review the manuscript titled "Artificial Neural Network Blockchain Techniques for Healthcare System: focusing on Personal Health Record". The general motivation of the manuscript is to find an application of AI and blockchain technologies in the modern healthcare sector. However, in the course of reading, I struggled to find what the authors were actually doing and ultimately failed to comprehend them.

Looking through the manuscript in general, I concluded that the paper is a joke. Let me share some clues:

Figs. 1, 2, 5, 6, and 7 throw a bunch of hot keywords at the reader without actually explaining or presenting something. Specifically, at the Fig.2, authors try to show that the outcome of the output layer of the neural network is “Intelligence”, and “Self organisation” - and none of it is even remotely true: neural networks are the method of _supervised_ learning, unable to any self-organisation and their abilities in pattern recognition are quite far from the common meaning of “intelligence”. Fig.6 is nothing more than a random collection of machine learning keywords.

Background knowledge section discusses the mechanistic basis of neural networks, then goes to deep learning, and then arrives to perceptron algorithm, which precedes the first two sections by decades and which relevance to the topic is questionable at best. This material (except a deep learning maybe) is covered in every single coursebook on machine learning. There is no background on the healthcare aspect of the problem.

Figures 8 - 10 are random snippets of code executing very mundane tasks.

Overall, I recommend the editor to reject this manuscript and have a little to recommend to authors.

Author Response

Comments and Suggestions for Authors

Thank you for the opportunity to review the manuscript titled "Artificial Neural Network Blockchain Techniques for Healthcare System: focusing on Personal Health Record". The general motivation of the manuscript is to find an application of AI and blockchain technologies in the modern healthcare sector. However, in the course of reading, I struggled to find what the authors were actually doing and ultimately failed to comprehend them.

Looking through the manuscript in general, I concluded that the paper is a joke. Let me share some clues:

Figs. 1, 2, 5, 6, and 7 throw a bunch of hot keywords at the reader without actually explaining or presenting something. Specifically, at the Fig.2, authors try to show that the outcome of the output layer of the neural network is “Intelligence”, and “Self organisation” - and none of it is even remotely true: neural networks are the method of _supervised_ learning, unable to any self-organisation and their abilities in pattern recognition are quite far from the common meaning of “intelligence”. Fig.6 is nothing more than a random collection of machine learning keywords.

Background knowledge section discusses the mechanistic basis of neural networks, then goes to deep learning, and then arrives to perceptron algorithm, which precedes the first two sections by decades and which relevance to the topic is questionable at best. This material (except a deep learning maybe) is covered in every single coursebook on machine learning. There is no background on the healthcare aspect of the problem.

Figures 8 - 10 are random snippets of code executing very mundane tasks.

Overall, I recommend the editor to reject this manuscript and have a little to recommend to authors.

Reply->

Thank you for your valuable comments. I revised the paper as per your recommendation. I also respect your opinion.

ADD)

Figure 1. Healthcare Verification Concepts with Blockchain and Artificial Intelligence

And this paper designs the image data of medical data EMR(Electric Medical Record), PACS(Picture Archiving Communication System), and OCS(Order Communication System),PHR(Personal Health Record)to match the actual various medical data. The reason is that medical data can only be seen by doctors now, but they use medical terminology. So in this paper, the first meaning is to eliminate forgery and falsification of medical data. And secondly, medical data is important personal information. And because it is sensitive information, these medical data are regulated very strongly in Europe as well as in Korea, which is called GDPR(General Data Protection Regulation). The purpose of this important medical data is to prevent forgery and to help personal medical data flow.

Figure 2. Concept of Artificial Intelligence Neural Network For Healthcare

Also, there is a lot of data using the Artistic Neural Network. And recently, medical data is being stored in many data warehouses. Therefore, medical imaging data is analyzed with blockchain using these data warehouse. It plays a role in verifying using blockchain agreement algorithm in the middle. This paper uses these blockchain verification algorithms and medical data uses artificial intelligence Neural Network. Therefore, it is a safer way for patients to determine their personal information's self-controlling rights. These EMR, PACS, OCS, and PHR data are highly sensitive data, so no errors should occur to validate medical data. To study algorithms to overcome these points.

Figure 5. Blockchain Algorithm For Healthcare

And blockchain has many algorithms. In particular, we want to apply the health information blockchain using Hyper POR algorithm, which is an agreement algorithm among public blockchain. The reason for using Hyper POR algorithm is that these medical data will become a Personal Health Record (PHR) in the future, requiring a large number of Translation Per Seconds (TPS) if many data are used. This is because many TPS performance data are required to authenticate these blockchain. Therefore, the health information blockchain needs a high-performance consensus algorithm. It also includes a smart contract function that automatically contracts. In this paper, a number of blockchain agreement algorithms, shading technology and smart contract technology are used to verify these health information data.

Figure 6. CNN Intelligent Agent Cloud Architecture to Increase Healthcare Data Readings

It presents an architecture that verifies medical data using artificial intelligence algorithms in multi-dimensional arrays. This works with the Hyper POR algorithm methodology, a blockchain agreement algorithm. Also, this data is what creates a decentralized system with shading technology. The CNN algorithm will also validate PACS(Picture Archiving Communication System), MRI(Magnetic Resonance Imaging) and CT(Computed Tomography) data. After verifying the original data, the data is then placed in the block of the blockchain, then smart contracts and distributed separately by consensus algorithms. It's an architecture for this. This paper validates these architectures and requires Cross Entropy Error, Mean Error, Mean Squared Error, Loss Function. It presents the first verification data based on the Neural network and the architecture based on the blockchain agreement algorithm after verification.

Figure 7. Error Backpropagation Framework For Healthcare

In addition, backpro placement is used to draw data architecures based on IFRS data. Also, artificial intelligence data is loaded into the cloud system to verify numerous medical data. In this paper, blockchain verification dataware house is created and these data are mounted on EHR, EMR, PACS, OCE data etc. And all medical data is mounted based on MRI, CT and X-RAY data. It is very important to verify the convergence of these data initially. Because it can be very wrong for inconsistent data to be stored in blockchain blocks. These data-based blockchain are loaded.

This source code is an example of the actual machine learning. It is difficult to disclose all sources in this paper. However, it is used as data to extract some source codes and verify them. These source codes receive variables as mnist import load_mnost. And the program used Python program. The source code is verified by receiving a random.choice. Contains procedures for verifying the necessary sources.

Reviewer 2 Report

The comments are attached.

Author Response

I recommend that manuscript should be rejected.

First of all, the manuscript fits the aims and scope for Electronics.

Secondly, comments are summarizing.

  1. Revisions. (1) Line 151-155 are verbose. (2) The authors shall cite correct and appropriate references in Section 2.. For example, Reference [4] and [5] don’t reflect terminology properly. Similar situations happens at Figure 3. in Section 2.2.. (3)Section2. shallberevisedfurtherandmorespecificonterminologyandmethodology. Istrongly suggest that entire Section 2. shall be rewritten. Too many pleonasms and verbose sentences are in Abstract.

Reply->

Thank you for your valuable comments. I revised the paper as per your recommendation. I also respect your opinion.

ADD)

Although the structure of the artificial neural network itself began by simulating the structure of the biological brain, there are structural differences other than the number of neurons. A typical cell has a working rate characteristic that, if it does not receive stimuli above the threshold, does not respond at all but reacts when it is stimulated above the threshold.

is change word

Artificial neural networks are neural networks that humans have and are used for deep learning. So I made it in deep learning like neuron's neural network.

2.1. Artificial Intelligence Neural Network

These problems can be solved by introducing a nonlinear active function, which allows nonlinear problems to be solved with artificial neural network models and generally leads to improvement with many layers. Moreover, for these nonlinear active functions, the local characteristics of the function generally change significantly if the input value is greater than a specific value, which is very similar to activating and responding when the cell is stimulated above a threshold; thus, these nonlinear functions are called active functions.

is delete

2.2. Deep Learning

Just doing the error subtraction from this will cause extreme variability, for which a function can be used to cushion it. A function that came out a long time ago is the Sigmoid function (the structure of the sigmoid function is a function that converges rapidly into zero as errors become smaller). Since the sigmoid function is a function that converges on zeros as previously stated, the deeper the layer is, the closer the gradient value is to zero, the more likely it can promote the problem of slope extinction. It can be solved using partially linear functions, such as the most commonly used RELU at present.

is delete

2.4. Blockchain

In other words, they have inherent limitations associated with operating principles. Exploitation is very difficult in some mature blockchain because the more unspecific nodes participate, the more expensive the hardware and electricity are in maintaining and controlling 51% of nodes. While control costs are a problem, attacking mature blockchain is meaningless unless it is a political maneuver as the chain-linked encryption system crashes even if the attack is successful. In the case of early e-money, where mining groups are small or not yet mature, there is sufficient attack that in theory cannot be prevented. In addition, 51% maintenance is necessary for the smooth control of the initial blockchain; thus, the moment 51% of the attack is launched, the cracker has the say.

is delete

2.5. Smart Healthcare

Personalized data may also help prescribe the side effects of individual drugs. In particular, the government is pushing for R&D policies related to artificial intelligence at the pan-governmental level as cheap and rapid medical services are required due to the aging population and the burden of medical expenses worldwide.

is delete

  1. Backgrounds. It is short of mathematical terminology and background for a SCI journal. I suggest that the authors shall address activation function (especially, the function you used) at least or maybe a mathematics section. Detail flow charts are required.

Reply->

Thank you for your valuable comments. I revised the paper as per your recommendation. I also respect your opinion.

3.4. CNN Intelligent Agent Cloud Architecture Flowchart

This paper has the stage of verifying CNN of blockchain-based artificial intelligence.

Figure 11. CNN Intelligent Agent Cloud Architecture Flowchart

Step 1) The CNN Intelligent Agent Cloud Architecture goes to Block.io. This is for data analysis. Data is collected and classified by type. It also refines data. Distribute final data.

Step 2) Take the blockchain algorithm that verifies these refined data. Blockchain uses Hyper POR for agreement algorithm. Hyper POR algorithm works by verifying Business Partner. Then set the Generations Block that verifies in the middle. After that, we add shading technology to do distributed computing.

Step 3) Verified data can be verified by CNN verification algorithm using this refined blockchain technique.In particular, EMR and PACS data are verified by ReLU, Learning Rate and Epoch verification algorithms.

Step 4) It is a step in verifying the collected data. Verification is made using the Stair Function Step Function, Sigmoid Function Sigmoid Fraction, Nonlinear Function, Multidimension Arrangement functions.

Step 5) Final Step 5 verifies verified data with probabilities. It verifies the convergence of the collected data and ultimately delivers the most error-prone EMR, PACS, and so on (see Figure 11).

III. Methodology. (1)The present study was designed to explore blockchain to ensure safe using certain data. In my opinion, the authors offer an interesting way to obtain subjects. However, the present manuscript lacks of comparison to other methodology. More specific description/comparison is suggested for Table 1.. (2) The performance/improvements on TPS(Line 612, 619 and 623) are fine.

Reply->

Thank you for your valuable comments. I revised the paper as per your recommendation. I also respect your opinion.

Table 1. Smart Healthcare for Hardware, Software, and Service

Items

Hardware

Software

Service

Purpose

Hardware system such as robot during artificial intelligence research

Software technology as key to the study of artificial intelligence

Personalized models found

Related research

Wearable devices, parts, devices, reagents, etc.

Providing medical healthcare content, communication network platform, medical information, exercise information, etc.

Genes, medical diagnosis services, genetic information

Speed

Measurements

High

Middle

Slow

Robot system for healthy strengthening

Personalized, integrated medical device services

Hardware and software mixed service required

Middleware

None

Middleware required

Needed

Technical technology

Blood sugar, blood pressure, ECG, activity measurement, chemical analysis, body fat analysis, medical sensors, field testing devices, band-necked implants

WebnisApp, nutrition management app, personal healthcare app

Personal health examination services, personal health records management systems, and healthcare services for the elderly

Artificial Intelligence Application Phase

Hardware Compute Platform Important

Software Logic Important

Need to develop a service model

Blockchain Application Phase

Total Level

Upper Level

Under Level

Effects of Consensus Algorithm

Middle Effect

High Effect

Low Effect

CNN Protocol Validation

Verification

Important

More Important

Non-Importnat

Reviewer 3 Report

In this paper, the authors describe an Artificial Neural Network (ANN) Blockchain system. The Transection Per Second (TPS) results and the UI design is well-presented.

1) The structure of this paper may need major revision. In the introduction, it would be better to include a figure to show exactly what a ANN Blockchain system is - what is stored on-chain, what is stored off-chain, how the ANN component interact with the Blockchain part, and how the User Interface interact with ANN/Blockchain parts? Then, would need to describe the state-of-the-art blockchain-based predictive modeling methods such as [1][2][3], and compare them with the proposed system to show what challenges need to be tackled and why the proposed ANN Blockchain system can improve them. Also, the discussion of basic concept of ANN and Blockchain can largely be replaced by citations to introductory papers and/or textbooks.

2) The blockchain platform used in the system is not clearly described (guessing that it might be Ethereum). Please explicitly describe it, and compare it with other platforms referring to studies such as [4][5][6].

3) Minor: please spell-out the abbreviations when they first appear (e.g., PHR, EHR, TPS, etc.).

[1] T.-T. Kuo, C.-N. Hsu, L. Ohno-Machado, paper presented at the ONC/NIST Use of Blockchain for Healthcare and Research Workshop, September 26, 2016 - September 27, 2016. Gaithersburg, Maryland, United States, 2016.
[2] X. Chen, J. Ji, C. Luo, W. Liao, P. Li, paper presented at the 2018 IEEE International Conference on Big Data (Big Data), December 10, 2018 - December 13, 2018. Seattle, WA, United States, 2018.
[3] T.-T. Kuo, R. A. Gabriel, L. Ohno-Machado, Fair compute loads enabled by blockchain: sharing models by alternating client and server roles. Journal of the American Medical Informatics Association (JAMIA). Edited by Suzanne Bakken. Published by Oxford University [4] Press, Kettering, Northants, UK. 26, 392-403 (2019). Weng J, Weng J, Zhang J, Li M, Zhang Y, Luo W. Deepchain: Auditable and privacy-preserving deep learning with blockchain-based incentive. IEEE Transactions on Dependable and Secure Computing. 2019 Nov 8.

[4] Chowdhury MJ, Ferdous MS, Biswas K, Chowdhury N, Kayes AS, Alazab M, Watters P. A Comparative Analysis of Distributed Ledger Technology Platforms. IEEE Access. 2019 Nov 15.

[5] Kuo TT, Zavaleta Rojas H, Ohno-Machado L. Comparison of blockchain platforms: a systematic review and healthcare examples. Journal of the American Medical Informatics Association. 2019 Mar 25;26(5):462-78.

[6] Macdonald M, Liu-Thorrold L, Julien R. The blockchain: a comparison of platforms and their uses beyond bitcoin. COMS4507-Adv. Computer and Network Security. 2017.

Thank you very much!

Author Response

Comments and Suggestions for Authors

In this paper, the authors describe an Artificial Neural Network (ANN) Blockchain system. The Transection Per Second (TPS) results and the UI design is well-presented.

1) The structure of this paper may need major revision. In the introduction, it would be better to include a figure to show exactly what a ANN Blockchain system is - what is stored on-chain, what is stored off-chain, how the ANN component interact with the Blockchain part, and how the User Interface interact with ANN/Blockchain parts? Then, would need to describe the state-of-the-art blockchain-based predictive modeling methods such as [1][2][3], and compare them with the proposed system to show what challenges need to be tackled and why the proposed ANN Blockchain system can improve them. Also, the discussion of basic concept of ANN and Blockchain can largely be replaced by citations to introductory papers and/or textbooks.

Reply->

Thank you for your valuable comments. I revised the paper as per your recommendation. I also respect your opinion.

I refer to the latest papers[1], [2], and [3].

It also requires multidisciplinary collaboration among computer scientists, engineers and data scientists as well as statisticians and other stakeholders to make the most of the potential of big data. It also calls for enhancing decision-making and service delivery by improving the big data processing, management and analysis infrastructure through large-scale investment and development by businesses and other organizations. Moreover, there is an urgent need to integrate new big data applications with existing application interfaces (APIs) such as structured query language (SQL) and R language for statistical computing. Several leading IT companies such as IBM, Oracle, Microsoft, SAP and HP have already invested more than $15 billion in big data systems. One of the challenges big data faces in enterprise and cloud infrastructure is the presence of tenants with different service-level agreements (SLA) requirements and various workloads that must be hosted on the same set of clusters. The initial solution to these challenges at the application level is to leverage distributed file systems to control data access and sharing within the cluster. At the infrastructure level, solutions such as virtual machines (VMs) or Linux containers dedicated to each application or tenant enabled the separation of allocated resources. Big data systems are also plagued by security, privacy and governance concerns. It also needs to investigate and optimize the improved energy-efficient processing and networking infrastructure for future big data as the growing computing demand for data volumes exceeds the capabilities of existing commercial infrastructure[1-3].

2) The blockchain platform used in the system is not clearly described (guessing that it might be Ethereum). Please explicitly describe it, and compare it with other platforms referring to studies such as [4][5][6].

Reply->

Thank you for your valuable comments. I revised the paper as per your recommendation. I also respect your opinion.

I quoted the latest papers [4], [5], and [6].

And the Ethereum system is right.

Ethereum is stored as additional ledger data called DLT system featuringaviatingmachine, along with additional ledger data called DLT system featuringaviatingmachine. Like Bitcoin, it offers similar functions. In addition, Ethereum's EVM allows smart contracts to be placed and executed in public books, enabling the creation of immutable computer logic. Smart contract introduction and execution of smart contracts require Etherithostore's accrualative authority by spending a certain amount of money called Etherrium crypto-currence. Once placed in the ledger, the simple model of smart contract execution is as follows. Smart contracts run with some input data through transactions. An EVM run uses input data to conclude a mart contract and generate output. This action changes the state of the EVM stored in the ledger along with the output data. The PoW Consensus algorithm ensures that the updated status is accurately recorded on all nodes in the network. The open ledger ensures that the transfer of currencies through transactions and changes in the status of EVMs are fully transparent and verified by all participants[4-5].

3) Minor: please spell-out the abbreviations when they first appear (e.g., PHR, EHR, TPS, etc.).

[1] T.-T. Kuo, C.-N. Hsu, L. Ohno-Machado, paper presented at the ONC/NIST Use of Blockchain for Healthcare and Research Workshop, September 26, 2016 - September 27, 2016. Gaithersburg, Maryland, United States, 2016.

[2] X. Chen, J. Ji, C. Luo, W. Liao, P. Li, paper presented at the 2018 IEEE International Conference on Big Data (Big Data), December 10, 2018 - December 13, 2018. Seattle, WA, United States, 2018.

[3] T.-T. Kuo, R. A. Gabriel, L. Ohno-Machado, Fair compute loads enabled by blockchain: sharing models by alternating client and server roles. Journal of the American Medical Informatics Association (JAMIA). Edited by Suzanne Bakken. Published by Oxford University [4] Press, Kettering, Northants, UK. 26, 392-403 (2019). Weng J, Weng J, Zhang J, Li M, Zhang Y, Luo W. Deepchain: Auditable and privacy-preserving deep learning with blockchain-based incentive. IEEE Transactions on Dependable and Secure Computing. 2019 Nov 8.

[4] Chowdhury MJ, Ferdous MS, Biswas K, Chowdhury N, Kayes AS, Alazab M, Watters P. A Comparative Analysis of Distributed Ledger Technology Platforms. IEEE Access. 2019 Nov 15.

[5] Kuo TT, Zavaleta Rojas H, Ohno-Machado L. Comparison of blockchain platforms: a systematic review and healthcare examples. Journal of the American Medical Informatics Association. 2019 Mar 25;26(5):462-78.

[6] Macdonald M, Liu-Thorrold L, Julien R. The blockchain: a comparison of platforms and their uses beyond bitcoin. COMS4507-Adv. Computer and Network Security. 2017.

Reply->

Thank you for your valuable comments. I revised the paper as per your recommendation. I also respect your opinion. I revice PHR, HER,TPS

PHR(Personal Health Record)

EHR(Electric Health Record)

TPS(Transaction Per Second)

EMR(Electric Medical Record)

PACS(Picture Archiving Communication System)

OCS(Order Communication System)

MRI(Magnetic Resonance Imaging)

CT(Computed Tomography)

GDPR(General Data Protection Regulation)

  1. Introduction

And this paper designs the image data of medical data EMR(Electric Medical Record), PACS(Picture Archiving Communication System), and OCS(Order Communication System),PHR(Personal Health Record)to match the actual various medical data. The reason is that medical data can only be seen by doctors now, but they use medical terminology. So in this paper, the first meaning is to eliminate forgery and falsification of medical data. And secondly, medical data is important personal information. And because it is sensitive information, these medical data are regulated very strongly in Europe as well as in Korea, which is called GDPR(General Data Protection Regulation). The purpose of this important medical data is to prevent forgery and to help personal medical data flow.

And I compared the latest paper.

  1. T.-T. Kuo, C.-N. Hsu, L. Ohno-Machado, paper presented at the ONC/NIST Use of Blockchain for Healthcare and Research Workshop, September 26, 2016 - September 27, 2016. Gaithersburg, Maryland, United States, 2016.
  2. X. Chen, J. Ji, C. Luo, W. Liao, P. Li, paper presented at the 2018 IEEE International Conference on Big Data (Big Data), December 10, 2018 - December 13, 2018. Seattle, WA, United States, 2018.
  3. T.-T. Kuo, R. A. Gabriel, L. Ohno-Machado, Fair compute loads enabled by blockchain: sharing models by alternating client and server roles. Journal of the American Medical Informatics Association (JAMIA). Edited by Suzanne Bakken. Published by Oxford University [4] Press, Kettering, Northants, UK. 26, 392-403 (2019). Weng J, Weng J, Zhang J, Li M, Zhang Y, Luo W. Deepchain: Auditable and privacy-preserving deep learning with blockchain-based incentive. IEEE Transactions on Dependable and Secure Computing. 2019 Nov 8.
  4. Chowdhury MJ, Ferdous MS, Biswas K, Chowdhury N, Kayes AS, Alazab M, Watters P. A Comparative Analysis of Distributed Ledger Technology Platforms. IEEE Access. 2019 Nov 15.
  5. Kuo TT, Zavaleta Rojas H, Ohno-Machado L. Comparison of blockchain platforms: a systematic review and healthcare examples. Journal of the American Medical Informatics Association. 2019 Mar 25;26(5):462-78.
  6. Macdonald M, Liu-Thorrold L, Julien R. The blockchain: a comparison of platforms and their uses beyond bitcoin. COMS4507-Adv. Computer and Network Security. 2017.

Round 2

Reviewer 2 Report

I believe the manuscript has been significantly improved and now suggest publication in Electronics.

Reviewer 3 Report

The authors have addressed all comments, thank you very much!